# Bridging Time and Linguistics: LLMs as Time Series Analyzer through Symbolization and Segmentation

**Jianyang Qin[1], Chaoyang Li[1,2], Jinhao Cui[1], Lingzhi Wang[1], Zhao Liu[2], Qing Liao[1,2*]**

[1]Harbin Institute of Technology (Shenzhen), Shenzhen, China
[2]Pengcheng Laboratory, Shenzhen, China
`{22b351005, 22b951022, cuijinhao}@stu.hit.edu.cn,`
`{liuzhao08}@pcl.ac.cn, {wanglingzhi, liaoqing}@hit.edu.cn`

## Abstract

Recent studies reveal that Large Language Models (LLMs) exhibit strong sequential reasoning capabilities, allowing them to replace specialized time-series models and serve as foundation models for complex time-series analysis. To activate the capabilities of LLMs for time-series tasks, numerous studies have attempted to bridge the gap between time series and linguistics by aligning textual representations with time-series patterns. However, it is a non-trivial endeavor to losslessly capture the infinite time-domain variability using natural language, leading to suboptimal alignment performance. Beyond representation, contextual differences, where semantics in time series are conveyed by consecutive points, unlike in text by individual tokens, are often overlooked by existing methods. To address these, we propose $S^2$TS-LLM, a simple yet effective framework to repurpose LLMs for universal time series analysis through the following two main paradigms: (i) a spectral symbolization paradigm transforms time series into frequency-domain representations characterized by a fixed number of components and prominent amplitudes, which enables a limited set of symbols to effectively abstract key frequency features; (ii) a contextual segmentation paradigm partitions the sequence into blocks based on temporal patterns and reassigns positional encodings accordingly, thereby mitigating the structural mismatch between time series and natural language. Together, these paradigms bootstrap the LLMs' perception of temporal patterns and structures, effectively bridging time series and linguistics. Extensive experiments show that $S^2$TS-LLM can serve as a powerful time series analyzer, outperforming state-of-the-art methods across time series tasks.

## 1 Introduction

Time series analysis [1] is widely applied across diverse domains such as transportation, energy, climate, finance, and healthcare, supporting a variety of tasks such as forecasting and classification. Owing to its practical applications, time series analysis has seen significant progress, giving rise to various specialized models tailored for certain tasks and datasets [2, 3, 4]. Recently, Large Language Models (LLMs) [5] exhibit remarkable reasoning performance, generalization ability, and multimodal knowledge, which are gradually replacing specialized models to serve as foundation models applied in various time series analysis tasks [6]. However, the inherent disparity between natural language and time series poses significant alignment challenges when leveraging LLMs for time series analysis.

The primary distinction between language and time series arises from their representational nature: natural language is composed of discrete tokens, whereas time series are continuously recorded

---

*Corresponding author

39th Conference on Neural Information Processing Systems (NeurIPS 2025).

in diverse formats such as univariate or multivariate [7]. To bridge this gap, a straightforward solution is to tokenize time series as sequences of digits [8], such as representing the value 1.23 as "1", ".", "2", "3", which is inefficient in representation and fails to capture temporal dependency inherent in time series. Alternatively, other approaches [9, 7] transform time series into customized word embeddings, such as "trend" and "peak", that are not only compatible with LLMs but also capable of capturing temporal patterns. Nevertheless, time series typically exhibit complex variations involving numerous patterns (e.g., rising, declining, stabilizing, etc.). As illustrated in Figure 1 (a), it remains a non-trivial endeavor to describe such intricate variations with concise language: the variation within the red

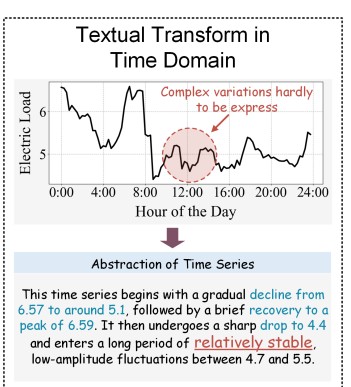 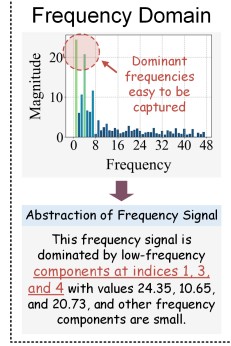

(a) Time-domain        (b) Freq-domain

Figure 1: Illustration of the challenge in representation alignment: time-domain sequences exhibit complex variations that are hard to express concisely, whereas frequency-domain signals are typically sparse and can be captured with a few textual tokens representing dominant components.

circle requires a detailed description of its upward and downward movements, rather than being simply summarized as "relatively stable". To address this, we propose a simple yet effective alignment paradigm, ***Spectral Symbolization***, to transform time-series patterns into textual embeddings. Technically, we apply the time-frequency transform [10] to efficiently compress time-domain sequences into compact frequency-domain representations. As depicted in the red circle in Figure 1 (b), the sparsity nature of the frequency domain enables the selection of a few dominant components to abstract temporal variations, mitigating the challenge of representing complex time-domain series accurately with natural language.

Beyond representation differences, the discrepancy in contextual structures [11] between time series and natural language is also significant and cannot be overlooked: each token in language carries semantic meaning [12] (depicted in Figure 2 (a)); in contrast, a single time point provides limited information, with meaningful patterns emerging only across consecutive observations [13] (as shown in Figure 2 (b)). Using the token-level position encoding of standard LLMs, which assigns sequential indices to individual time points, would disrupt trend continuity (e.g., downward trend) across consecutive points, impairing the model's awareness of temporal context. Upon this observation, we go beyond previous position processing and propose a ***Contextual Segmentation*** paradigm to align the contextual structures between natural language and time series. Specifically, we segment a sequence of time series into several blocks based on the semantic meaning between time points. As illustrated in Figure 2 (b), time points within the same block, sharing similar trend semantics, are assigned to the same positional index, allowing the LLMs to model temporal trend continuity.

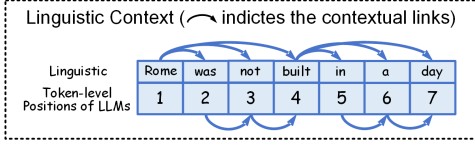

(a) Linguistic Context

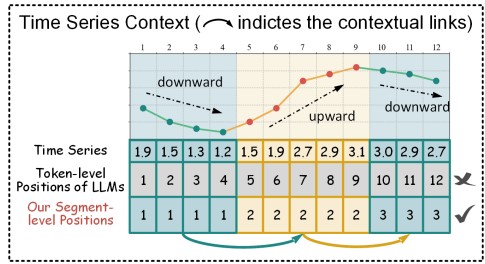

(b) Time Series Context

Figure 2: Illustration of the challenge in context alignment: standard LLMs assign positions sequentially to each token to capture context, whereas time series are better encoded by reassigning shared positions to correlated points to preserve trend continuity.

Building on these insights, we propose a novel framework, $S^2$**TS-LLM**, that repurposes the **LLM**s to align **T**ime **S**eries with the representational space and contextual structures of natural language through spectral **S**ymbolization and contextual **S**egmentation. Through these modules, we transform the time series into symbolic abstractions and segment-level

positions, which are combined with the raw sequence to form a more informative input for LLMs. The temporal and textual outputs of LLMs are subsequently fused to produce predictions for downstream tasks. Our comprehensive evaluation demonstrates that $S^2$TS-LLM can serve as an effective time series analyzer, providing versatility for diverse time-series analysis tasks, such as long-term/short-term forecasting, few-shot/zero-shot learning, and classification. The main contributions of this paper are summarized as follows:

- We propose $S^2$TS-LLM, a framework that repurposes LLMs as a foundational time series analyzer by bridging the representational and contextual gaps between time series and natural language.

- The proposed spectral symbolization and contextual segmentation paradigms endow time series with frequency-based textual abstractions and language-like structural positions, bootstrapping the LLMs' comprehension of temporal variations and trend continuity.

- Extensive experiments on twenty-three datasets show that $S^2$TS-LLM can achieve performance that is comparable to or surpasses state-of-the-art methods across five major time series tasks.

## 2   Related Works

**Time Series Analysis.** Time series analysis, as a fundamental task for scientific and industrial applications, has undergone a significant revolution with the advancement of deep learning [14]. Conventional time-series models like ARIMA [15] were built upon statistical characteristics of the data, which are crafted for specific tasks. With the advent of deep learning, neural networks, such as RNNs [2], TCNs [16], MLPs [17], and Transformers [18, 19], offered task-agnostic solutions by mining the dynamic variations and long-term dependencies of time series. While these models have achieved considerable improvements, they lack generalizability across diverse time-series domains [20]. Inspired by the success of pre-training techniques in CV and NLP, recent work has explored pre-trained models for time series [21], aiming to enhance models' representation and transfer abilities across diverse domains. Although pre-training can improve models' versatility, such models remain restricted to the time-series modality, lacking sufficient data to expand their capacity and to enrich their knowledge [22]. Consequently, they fall short in handling complex real-world analytical tasks.

**Large Language Models for Time Series Analysis.** Compared to pre-trained models, LLMs, which acquire vast knowledge from diverse large-scale datasets, have demonstrated potential as universal analyzers applicable not only to CV and NLP but also to time series tasks [23]. Recently, a surge of LLM-based methods tailored for time series analysis has emerged [24, 25], but they inevitably face the challenge of cross-modal adaptation—transferring internal knowledge to align with time series representations [26]. Early attempts addressed this via value-to-digit conversion [23], prompt engineering [27, 28], or fine-tuning [22], yet these simplistic approaches struggle to capture temporal dependencies. Therefore, more recent work focuses on aligning textual and temporal semantics. For example, Time-LLM [9] reprograms time series into textual prototypes; CALF [5] reduces the distributional discrepancy between time-series and textual modalities in both the feature and output spaces; $S^2$IP-LLM [7] aligns decomposed components with textual anchors in a joint space; and TEST [29] applies contrastive learning to implicitly map text embeddings to temporal patterns. However, these methods rely on an unrealistic assumption that a limited set of textual tokens can faithfully represent complex and variable time-domain dynamics. Moreover, they often overlook fundamental differences in contextual structure between language and time series. To this end, this work aims to align time-series and language in the frequency domain, while reconstructing temporal contexts to better resemble textual structures.

## 3   Methodology

Given a sequence of textual tokens $\boldsymbol{X} = \{x_i\}_{i=1}^{l-1}$, LLMs typically encode tokens via a tokenizer $\psi(\cdot)$ and assign sequential positional indices $\boldsymbol{P} = \{p_i\}_{i=1}^{l-1}$ for tokens to generate next output tokens $\boldsymbol{Y} = \{y_i\}_{i=l}^{L}$ as follows,

$$\{\boldsymbol{y}_i\}_{i=l}^{L} = \textbf{LLMs}\left(\psi\left(\{\boldsymbol{x}_i\}_{i=1}^{l-1}\right), \{p_i\}_{i=1}^{l-1}\right), \quad \text{where} \quad p_i = i \,\forall i \in \{1, \ldots, l-1\}, \qquad (1)$$

where $\psi(\cdot)$ transforms tokens into $d$-dimension representations. $p_i$ indicates the position of token $x_i$ in a sequence, allowing LLMs to capture the token's contextual dependence. When applying LLMs

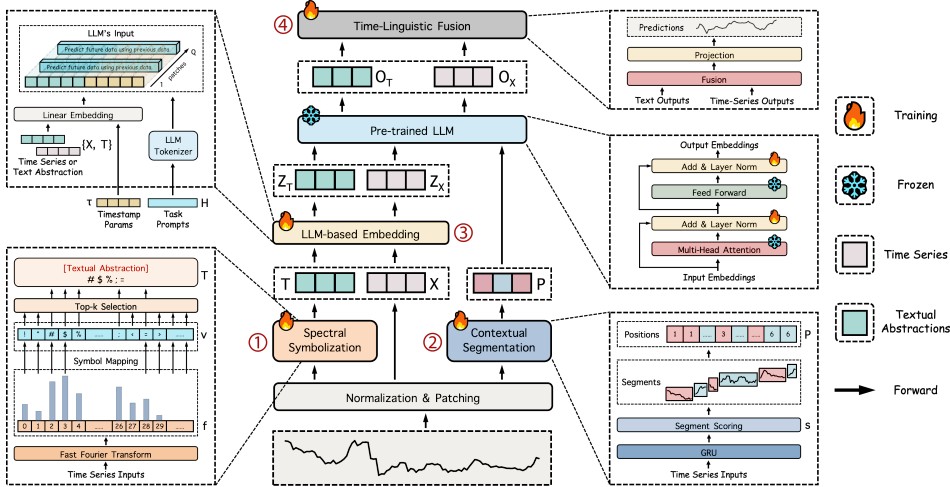

Figure 3: The framework of S$^2$TS-LLM. After processing the input time series through ① spectral symbolization and ② contextual segmentation, we obtain time-aware textual abstractions and remapped positions. ③ These, together with the time series, are embedded, packed with prompts, and fed into the pre-trained LLM. ④ The LLM outputs are fused and projected to generate predictions.

to time series analysis, the inherent gaps in representation space and contextual structure between time series and natural language hinder the models' capacity for temporal reasoning. To unlock the potential of LLMs for time series tasks, we focus on bridging time series and linguistics by (i) learning textual representations to align with time series patterns and (ii) reassigning the positional indices to alleviate the contextual mismatches.

## 3.1 Overview

As depicted in Figure 3, our framework encompasses four core components: (1) spectral symbolization, (2) contextual segmentation, (3) LLM-based embedding, and (4) time-linguistic fusion. Given a sequence of observations $\boldsymbol{X} \in \mathbb{R}^{N \times L}$, where $\boldsymbol{X}$ consists of $N$ univariate time series over $L$ time steps. Following conventional LLM-based approaches [22], we initially separate multivariate time series into $N$ univariate series, where the $n$-th univariate time series is denoted as $\boldsymbol{X}^{(n)} \in \mathbb{R}^{1 \times L}$. Each univariate time series is partitioned independently into $Q = \lfloor \frac{L-M}{R} \rfloor + 1$ patches, denoted as $\hat{\boldsymbol{X}}^{(n)} \in \mathbb{R}^{Q \times M}$, using a patch size of $M$ and a stride of $R$. Subsequently, these patches undergo a spectral symbolization to craft a textual abstraction $\boldsymbol{T}^{(n)}$ for representing temporal patterns, and a contextual segmentation to reassign positional indices $\boldsymbol{P}^{(n)}$ for capturing temporal context. Upon feedforwarding time series, textual abstractions and positional indices through a pre-trained LLM, we fuse the LLM's time-linguistic outputs and project them to derive the predictions $\hat{\boldsymbol{Y}}^{(n)}$ for downstream tasks such as forecasting, classification, etc., with the overall objective to minimize the errors between ground truth $\boldsymbol{Y}$ and predictions.

## 3.2 Spectral Symbolization

To bridge the representation gap between time series and linguistics, a common strategy is aligning time series with existing vocabulary via transformations, such as converting time series into word embeddings [24, 23] or prototypes [9, 7]. However, these transformations aim to textually represent temporal sequence in a time domain rather than frequency domain, yet they struggle to losslessly describe complex variations of time series. For example, a sequence $[66, 45, 24, 32, 27, 33, 16, 8]$ could be transformed into a textual abstraction like "*The series starts at 66 and drops to 24 over the first three points. After reaching 24, it briefly fluctuates between 32, 27, and 33, then falls again to 16 and finally 8*". While accurate, this lossless abstraction is far longer than the original sequence, reducing the efficiency of LLM learning and limiting practical applicability. In practice, a common alternative is to simplify temporal dynamics by tokenizing the same sequence merely as "*decreasing from 66 to 8*". While this achieves efficiency, it overlooks critical local variations (e.g.,

the intermediate fluctuations $32 \to 27 \to 33$). As such, how to textually abstract time series in a brief yet precise manner—thereby enhancing LLMs' temporal comprehension—remains an open issue.

To achieve this, we propose a spectral symbolization paradigm that transforms frequency-domain patterns of time series $\hat{X}^{(n)} \in \mathbb{R}^{Q \times M}$ into symbolic sequences, rather than modeling time-domain features. Technically, a time series can be expressed as a superposition of sine waves characterized by various frequencies through Fast Fourier Transform (FFT) [10],

$$\left[f_{q,1}, f_{q,2}, \cdots, f_{q,\lfloor \frac{M}{2} \rfloor}\right] = \text{FFT}\left(\hat{X}_q^{(n)}\right), \tag{2}$$

where $\hat{X}_q^{(n)}$ is the $q$-th patch of time series and $f_{q,*}$ denotes a specific frequency. $\lfloor \frac{M}{2} \rfloor$ is the number of frequencies, equal to half of the dimensions $M$ because of the symmetry of FFT. Compared to time-domain data, frequency-domain signals are typically sparse and dominated by a few components, which helps reduce noise caused by complex temporal variations [13]. The compact nature of frequency allows us to use a finite and fixed set of textual tokens for time-series representation,

$$\left[v_1, v_2, \cdots, v_{\lfloor \frac{M}{2} \rfloor}\right] \longleftarrow \left[f_{q,1}, f_{q,2}, \cdots, f_{q,\lfloor \frac{M}{2} \rfloor}\right], \tag{3}$$

where $v_m$ is the $m$-th textual token corresponding to $f_{q,m}$. To distinguish time series representation between patches, we further identify salient textual tokens from $\left[v_1, \cdots, v_{\lfloor \frac{M}{2} \rfloor}\right]$ for each patch $\hat{X}_q^{(n)}$ to describe its unique patterns. Specifically, we select the textual tokens corresponding to the top-$k$ frequencies with the highest amplitudes to form an abstraction of the time series, denoted as $T_q^{(n)} = \{v_i\}_{i=1}^k$,

$$\{v_1, v_2, \cdots, v_k\} \underset{k \in \{1, \cdots, \lfloor \frac{M}{2} \rfloor\}}{\longleftarrow} \{f_{q,1}, f_{q,2}, \cdots, f_{q,k}\} = \underset{k \in \{1, \cdots, \lfloor \frac{M}{2} \rfloor\}}{\arg \text{Topk}} \left(\text{Amp}\left(\text{FFT}\left(\hat{X}_q^{(n)}\right)\right)\right) \tag{4}$$

where $\text{Amp}(\cdot)$ represents the calculation of amplitude values. $\{f_{q,1}, f_{q,2}, \cdots, f_{q,k}\}$ are top-$k$ frequency components selected from the $q$-th data patch. Although these top-$k$ frequencies carry relatively compact information, they are capable of capturing the global patterns of a time series[30, 31, 32], supporting an effective characterization of the sequence. For instance, given the same sequence $[66, 45, 24, 32, 27, 33, 16, 8]$ mentioned above, using the top-2 frequency components can reconstruct a sequence $[57, 36, 14, 33, 32, 46, 23, 10]$ that preserves the original trend of "decreasing→fluctuating→decreasing".

For mapping a frequency $f_{q,k}$ to a token $v_k$, one straightforward approach is to assign predefined textual vocabulary (such as "trend" and "peak") to each frequency based on the datasets and tasks. However, this requires labor-intensive manual selection and lacks generalizability across different settings. As an alternative, we choose the first $\lfloor \frac{M}{2} \rfloor$ ASCII symbols (e.g., "@", "!", or "+") to describe corresponding $\lfloor \frac{M}{2} \rfloor$ frequencies. Although these symbols do not carry explicit semantics, LLMs can use the statistical properties learned during pretraining [33, 34, 35] to interpret them like a foreign language, associating different symbol combinations with distinguishable spectral patterns. Consequently, the mapped top-$k$ textual tokens $\{v_1, v_2, \cdots, v_k\}$ provides concise yet unambiguous cues that enable LLMs distinguish global time-series trends such as periodicity, volatility, or smoothness. In this paper, we implement the model using the ASCII symbol mapping scheme, while the implementation details and results of the predefined vocabulary mapping are additionally reported in the Appendix E.6.

### 3.3 Contextual Segmentation

Due to the contextual discrepancy between time series and natural language, the token-level position encoding assigns sequential indices to time-series patches (as shown in Eq. 1), which hinders LLMs from capturing temporal trend continuity, such as downward and upward movements. To address this, we propose a contextual segmentation paradigm that dynamically partitions a sequence of $Q$ patches into several blocks and assigns a positional index to each block. Regarding an input time series $\hat{X}^{(n)} = \left[\hat{x}_1^{(n)}, \cdots, \hat{x}_Q^{(n)}\right]$, we first pass it through a GRU to get hidden embeddings $\left\{c_q^{(n)}\right\}_{q=1}^Q$, and then calculate their segment score $s$ through $\ell_2$-distance,

$$\left\{c_q^{(n)}\right\}_{q=1}^Q = \text{GRU}\left(\left\{\hat{x}_q^{(n)}\right\}_{q=1}^Q\right), \quad s(i,j) = \|c_i^{(n)} - c_j^{(n)}\|_2. \tag{5}$$

Intuitively, the score $s(i,j)$ determines whether the patches $i$ and $j$ should belong to the same block: if $s(i,j) < \gamma$, they are grouped together; otherwise, they are separated. After obtaining $B$ segments, we assign sequential positional indices $\boldsymbol{P}^{(n)}$ from 1 to $B$ to the resulting segments. Here, we provide an intuitive example to illustrate the difference between our segment-level position and token-level position encoding of standard LLMs: given a time series with 6 patches, conventional token-level encoding assigns sequential indices $[1, 2, 3, 4, 5, 6]$, whereas our approach may assign grouped indices like $[1, 1, 1, 2, 2, 3]$ to better reflect structural continuity. These segment-level positions are subsequently fed into the LLMs to enhance the model's awareness of temporal context. Notably, segmentation is used solely for reassigning positions and does not alter the input time series itself.

## 3.4 LLM-based Embedding

Given the time series $\hat{\boldsymbol{X}}^{(n)}$ and learned textual abstraction $\boldsymbol{T}^{(n)}$, we embed them using a simple linear layer $\phi(\cdot)$ and concatenate them with a learnable timestamp parameter $\boldsymbol{\tau}$,

$$\boldsymbol{E}_X^{(n)} = \phi(\boldsymbol{X}^{(n)}) \oplus \boldsymbol{\tau}, \quad \boldsymbol{E}_T^{(n)} = \phi(\boldsymbol{T}^{(n)}) \oplus \boldsymbol{\tau}. \tag{6}$$

Then, inspired by success of prompt engineering [36], we augment the embeddings $\boldsymbol{E}_X^{(n)}$ and $\boldsymbol{E}_T^{(n)}$ with task-specific prompts $\boldsymbol{H}$ (e.g., "*Predict future sequences using previous data*" for forecasting task) as LLMs' input tokens $\boldsymbol{Z}_X^{(n)}$ and $\boldsymbol{Z}_T^{(n)}$. Regarding that an embedding $\boldsymbol{E}_* \in \{\boldsymbol{E}_X, \boldsymbol{E}_T\} = [\boldsymbol{e}_{*,1}, \boldsymbol{e}_{*,2}, \cdots, \boldsymbol{e}_{*,B}]$ is divided into $B$ blocks through contextual segmentation and a prompt $\boldsymbol{H} = [h_1, h_2, \cdots, h_W]$ has $W$ words, we split the embeddings into $G$ slices and insert the prompt at the end of each slice as follows,

$$\boldsymbol{Z}_*^{(n)} = \left\{ \boldsymbol{e}_{*,g \times U+1}^{(n)}, \cdots, \boldsymbol{e}_{*,g \times U+U}^{(n)}, h_1, \cdots, h_W \mid g \in [0, 1, \cdots, G-1], U = \lfloor \frac{B}{G} \rfloor \right\}. \tag{7}$$

The token sequences $\boldsymbol{Z}_X^{(n)}$ and $\boldsymbol{Z}_X^{(n)}$ are processed by a shared backbone LLM with segment-level positional indices $\boldsymbol{P}^{(n)}$, yielding the output representations $\boldsymbol{O}_X^{(n)}$ and $\boldsymbol{O}_T^{(n)}$, respectively. Following common fine-tuning practices for LLMs [22], we freeze all components of the LLMs except for Layer Normalization (LN) layers, allowing time series adaptation.

## 3.5 Time-Linguistic Fusion

As the textual output $\boldsymbol{O}_T^{(n)}$ represent frequency-domain patterns distinct from the time-domain output $\boldsymbol{O}_X^{(n)}$, we treat $\boldsymbol{O}_T^{(n)}$ as a complementary guide and dynamically integrate it with $\boldsymbol{O}_X^{(n)}$ via a gated residual update, ensuring that predictions remain in the time domain:

$$\boldsymbol{O}^{(n)} = \boldsymbol{O}_X^{(n)} + \boldsymbol{A}^{(n)} \odot \sigma(\boldsymbol{O}_T^{(n)}) \odot \boldsymbol{O}_X^{(n)}, \quad \boldsymbol{A}^{(n)} = \frac{\boldsymbol{O}_T^{(n)} \boldsymbol{O}_X^{(n)}}{\|\boldsymbol{O}_T^{(n)}\| \|\boldsymbol{O}_X^{(n)}\|}. \tag{8}$$

Here, $\sigma(\cdot)$ denotes a sigmoid activation that nonlinearly transforms spectral features, serving as a gating signal [37] to leverage frequency information in controlling which components of the time-domain signal are preserved or suppressed. The cosine similarity $\boldsymbol{A}^{(n)}$ further modulates the fusion, emphasizing time-domain components that are semantically aligned with frequency-domain patterns. Finally, the fused outputs $\boldsymbol{O}^{(n)}$ are processed through prefix truncation [9], flattening, and a linear projection to produce the final prediction $\hat{\boldsymbol{Y}}^{(n)}$.

## 4 Experiments

To evaluate the model's effectiveness, we adhere to the experimental settings in [22] and conduct extensive experiments on five mainstream analysis tasks, including long/short-term forecasting, few-shot/zero-shot learning, and time-series classification. As shown in Figure 4, experimental results demonstrate that S$^2$TS-LLM[1] achieves performance comparable to or surpassing state-of-the-art models across diverse downstream tasks.

---
[1]https://github.com/JianyangQin/S2TS-LLM

**Baselines.** We compare our method against a series of well-known and advanced baselines, including LLM-based models (S$^2$IP-LLM [7], Time-LLM [9], GPT4TS [22]), Transformer-based models (TimeXer [38], iTransformer [39], PatchTST [40]), MLP-based models (TimeMixer [41], RLinear [17]), and a CNN-based model (TimesNet [13]). Notably, we uniformly use GPT-2 [42] as the backbone of all LLM-based models for fair comparison. Besides, a broader set of models is further employed for classification comparison. More experimental details are shown in Appendix B.

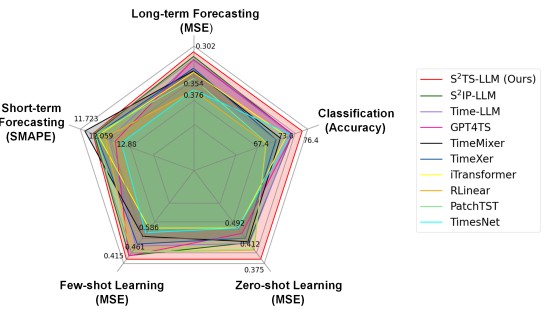

Figure 4: Overall Model Comparison.

## 4.1 Long-term Forecasting

**Setups.** We perform long-term forecasting on Weather, Electricity (ECL), Traffic, ETTh1, ETTh2, ETTm1, and ETTm2 datasets [13]. Following the experimental settings commonly used in LLM-based methods [22], the input time series length $L$ is set to 512, and the forecasting performance is evaluated across four prediction horizons {96, 192, 336, 720}, using Mean Squared Error (MSE) and Mean Absolute Error (MAE) as evaluation metrics.

**Results.** A summary of the forecasting results is presented in Table 1. Due to the expressive power of LLMs, LLM-based models, including our method, S$^2$IP-LLM, Time-LLM, and GPT4TS, generally outperform conventional baselines, highlighting the potential of LLMs in time series reasoning. Notably, while our method shares a similar goal of aligning time series with LLMs' textual representations as S$^2$IP-LLM and Time-LLM, our method consistently outperforms them. These improvements are mainly attributed to the following factors: (1) the spectral symbolization paradigm is capable of capturing frequency patterns, mitigating the difficulty of representing complex time-domain dynamics with discrete tokens; and (2) we further align contextual structures between time series and language, enhancing the LLMs' comprehension of sequential dependencies.

## 4.2 Short-term Forecasting

**Setups.** We conduct short-term forecasting using the M4 dataset [43], which comprises 100,000 time series sampled at various frequencies including yearly, quarterly, monthly, weekly, daily, and hourly. In this case, prediction horizons range from 6 to 48, with input lengths set to twice the prediction horizon. For evaluation, we adopt Symmetric Mean Absolute Percentage Error (SMAPE), Mean Absolute Scaled Error (MASE), and Overall Weighted Average (OWA) as metrics.

**Results.** As shown in Table 2, conventional specialized models, such as TimeMixer and PatchTST exhibit more impressive performance over LLM-based methods in short-term forecasting. A plausible explanation is that short input lengths constrain the statistical inference capabilities of LLMs. For example, the yearly subset of the M4 dataset provides only 12 time steps of data as input, offering insufficient long-term dependency; thus, LLMs are prone to overfit local noise rather than capture global statistical patterns. Nevertheless, our S$^2$TS-LLM still achieves competitive suboptimal performance, benefiting from its effective spectral representation and contextual learning capabilities.

Table 1: Long-term forecasting results. All the results are averaged from four different prediction horizons {96, 192, 336, 720}. Red denotes the best performance, while Blue indicates the second-best. Detailed results are provided in Appendix E.1.

| Methods | S$^2$TS-LLM (Ours) | | S$^2$ IP-LLM [7] | | Time-LLM [9] | | GPT4TS [22] | | TimeMixer [41] | | TimeXer [38] | | iTransformer [39] | | RLinear [17] | | PatchTST [40] | | TimesNet [13] | |
|---|---|---|---|---|---|---|---|---|---|---|---|---|---|---|---|---|---|---|---|---|
| Metric | MSE↓ | MAE↓ | MSE↓ | MAE↓ | MSE↓ | MAE↓ | MSE↓ | MAE↓ | MSE↓ | MAE↓ | MSE↓ | MAE↓ | MSE↓ | MAE↓ | MSE↓ | MAE↓ | MSE↓ | MAE↓ | MSE↓ | MAE↓ |
| Weather | 0.225 | 0.265 | 0.228 | 0.265 | 0.237 | 0.269 | 0.237 | 0.270 | 0.240 | 0.271 | 0.241 | 0.271 | 0.258 | 0.278 | 0.272 | 0.291 | 0.259 | 0.281 | 0.259 | 0.287 |
| ECL | 0.164 | 0.256 | 0.166 | 0.262 | 0.167 | 0.264 | 0.167 | 0.263 | 0.182 | 0.272 | 0.171 | 0.270 | 0.178 | 0.270 | 0.219 | 0.298 | 0.216 | 0.304 | 0.192 | 0.295 |
| Traffic | 0.396 | 0.278 | 0.405 | 0.286 | 0.407 | 0.289 | 0.414 | 0.294 | 0.484 | 0.297 | 0.466 | 0.287 | 0.428 | 0.282 | 0.626 | 0.378 | 0.481 | 0.304 | 0.620 | 0.336 |
| ETTh1 | 0.404 | 0.429 | 0.418 | 0.436 | 0.426 | 0.435 | 0.427 | 0.426 | 0.447 | 0.440 | 0.437 | 0.437 | 0.454 | 0.447 | 0.446 | 0.434 | 0.469 | 0.454 | 0.458 | 0.450 |
| ETTh2 | 0.321 | 0.376 | 0.355 | 0.399 | 0.361 | 0.398 | 0.361 | 0.398 | 0.354 | 0.394 | 0.367 | 0.396 | 0.383 | 0.407 | 0.374 | 0.398 | 0.387 | 0.407 | 0.414 | 0.427 |
| ETTm1 | 0.344 | 0.380 | 0.346 | 0.382 | 0.354 | 0.384 | 0.352 | 0.383 | 0.381 | 0.395 | 0.382 | 0.397 | 0.407 | 0.410 | 0.414 | 0.407 | 0.387 | 0.400 | 0.400 | 0.406 |
| ETTm2 | 0.258 | 0.318 | 0.262 | 0.326 | 0.275 | 0.334 | 0.266 | 0.326 | 0.275 | 0.323 | 0.274 | 0.322 | 0.288 | 0.332 | 0.286 | 0.327 | 0.281 | 0.326 | 0.291 | 0.333 |
| Average | 0.302 | 0.329 | 0.311 | 0.337 | 0.318 | 0.339 | 0.317 | 0.337 | 0.339 | 0.342 | 0.334 | 0.340 | 0.342 | 0.347 | 0.377 | 0.362 | 0.353 | 0.352 | 0.376 | 0.362 |

Table 2: Short-term forecasting results. The SMAPE, MASE, OWA results are weighted averaged across all M4 datasets, with prediction horizons ranging from [6, 48]. Red denotes the best performance, while Blue indicates the second-best. Detailed results are provided in Appendix E.2.

| Methods | S²TS-LLM (Ours) | S²IP-LLM [7] | Time-LLM [9] | GPT4TS [22] | TimeMixer [41] | TimeXer [38] | iTransformer [39] | RLinear [17] | PatchTST [40] | TimesNet [13] |
|---|---|---|---|---|---|---|---|---|---|---|
| SMAPE ↓ | 11.979 | 12.021 | 12.494 | 12.690 | 11.723 | 12.082 | 12.142 | 12.363 | 12.059 | 12.880 |
| MASE ↓ | 1.610 | 1.612 | 1.731 | 1.808 | 1.559 | 1.621 | 1.631 | 1.662 | 1.623 | 1.836 |
| OWA ↓ | 0.862 | 0.857 | 0.913 | 0.940 | 0.840 | 0.874 | 0.869 | 0.890 | 0.869 | 0.955 |

Table 3: Few-shot learning results on 5% and 10% training data. Detailed results are provided in Appendix E.3.

| Volume | Methods | S²TS-LLM (Ours) | | S²IP-LLM [7] | | Time-LLM [9] | | GPT4TS [22] | | TimeMixer [41] | | TimeXer [38] | | iTransformer [39] | | RLinear [17] | | PatchTST [40] | | TimesNet [13] | |
|---|---|---|---|---|---|---|---|---|---|---|---|---|---|---|---|---|---|---|---|---|---|
| | Metric | MSE ↓ | MAE ↓ | MSE ↓ | MAE ↓ | MSE ↓ | MAE ↓ | MSE ↓ | MAE ↓ | MSE ↓ | MAE ↓ | MSE ↓ | MAE ↓ | MSE ↓ | MAE ↓ | MSE ↓ | MAE ↓ | MSE ↓ | MAE ↓ | MSE ↓ | MAE ↓ |
| 5% | ETTh1 | 0.580 | 0.503 | 0.650 | 0.550 | 0.648 | 0.549 | 0.681 | 0.560 | 0.726 | 0.584 | 0.718 | 0.580 | 1.070 | 0.710 | 0.685 | 0.559 | 0.695 | 0.569 | 0.925 | 0.647 |
| | ETTh2 | 0.360 | 0.394 | 0.380 | 0.413 | 0.398 | 0.426 | 0.400 | 0.433 | 0.437 | 0.437 | 0.517 | 0.472 | 0.488 | 0.475 | 0.462 | 0.443 | 0.439 | 0.448 | 0.463 | 0.454 |
| | ETTm1 | 0.474 | 0.447 | 0.455 | 0.446 | 0.477 | 0.451 | 0.472 | 0.450 | 0.657 | 0.533 | 0.619 | 0.521 | 0.784 | 0.596 | 0.526 | 0.477 | 0.526 | 0.476 | 0.717 | 0.561 |
| | ETTm2 | 0.293 | 0.338 | 0.296 | 0.342 | 0.307 | 0.348 | 0.308 | 0.346 | 0.324 | 0.357 | 0.320 | 0.355 | 0.356 | 0.388 | 0.311 | 0.345 | 0.314 | 0.352 | 0.344 | 0.372 |
| | Average | 0.427 | 0.421 | 0.445 | 0.438 | 0.458 | 0.444 | 0.465 | 0.447 | 0.605 | 0.511 | 0.606 | 0.509 | 0.675 | 0.542 | 0.496 | 0.456 | 0.494 | 0.461 | 0.612 | 0.509 |
| 10% | ETTh1 | 0.543 | 0.493 | 0.593 | 0.529 | 0.785 | 0.553 | 0.590 | 0.525 | 0.700 | 0.568 | 0.685 | 0.552 | 0.910 | 0.860 | 0.625 | 0.532 | 0.633 | 0.542 | 0.869 | 0.628 |
| | ETTh2 | 0.372 | 0.402 | 0.419 | 0.439 | 0.424 | 0.441 | 0.397 | 0.421 | 0.456 | 0.446 | 0.474 | 0.451 | 0.489 | 0.483 | 0.463 | 0.443 | 0.415 | 0.431 | 0.479 | 0.465 |
| | ETTm1 | 0.465 | 0.436 | 0.455 | 0.435 | 0.487 | 0.461 | 0.464 | 0.441 | 0.598 | 0.502 | 0.557 | 0.481 | 0.728 | 0.565 | 0.455 | 0.439 | 0.501 | 0.466 | 0.677 | 0.537 |
| | ETTm2 | 0.279 | 0.327 | 0.284 | 0.332 | 0.305 | 0.344 | 0.293 | 0.335 | 0.301 | 0.336 | 0.306 | 0.343 | 0.336 | 0.373 | 0.295 | 0.330 | 0.296 | 0.343 | 0.320 | 0.353 |
| | Average | 0.415 | 0.415 | 0.438 | 0.434 | 0.500 | 0.450 | 0.436 | 0.431 | 0.562 | 0.478 | 0.513 | 0.460 | 0.616 | 0.570 | 0.460 | 0.436 | 0.461 | 0.446 | 0.586 | 0.496 |

Table 4: Zero-shot learning results from a source dataset to an unseen target dataset (e.g., ETTh1 → ETTh2). Detailed results are provided in Appendix E.4.

| Methods | S²TS-LLM (Ours) | | S² IP-LLM [7] | | Time-LLM [9] | | GPT4TS [22] | | TimeMixer [41] | | TimeXer [38] | | iTransformer [39] | | RLinear [17] | | PatchTST [40] | | TimesNet [13] | |
|---|---|---|---|---|---|---|---|---|---|---|---|---|---|---|---|---|---|---|---|---|
| Metric | MSE ↓ | MAE ↓ | MSE ↓ | MAE ↓ | MSE ↓ | MAE ↓ | MSE ↓ | MAE ↓ | MSE ↓ | MAE ↓ | MSE ↓ | MAE ↓ | MSE ↓ | MAE ↓ | MSE ↓ | MAE ↓ | MSE ↓ | MAE ↓ | MSE ↓ | MAE ↓ |
| ETTh1 → ETTh2 | 0.341 | 0.387 | 0.403 | 0.417 | 0.384 | 0.409 | 0.406 | 0.422 | 0.374 | 0.397 | 0.390 | 0.407 | 0.457 | 0.455 | 0.380 | 0.401 | 0.380 | 0.405 | 0.421 | 0.431 |
| ETTh1 → ETTm2 | 0.294 | 0.350 | 0.325 | 0.360 | 0.317 | 0.370 | 0.325 | 0.363 | 0.321 | 0.362 | 0.330 | 0.366 | 0.360 | 0.390 | 0.316 | 0.356 | 0.314 | 0.360 | 0.327 | 0.361 |
| ETTh2 → ETTh1 | 0.550 | 0.513 | 0.669 | 0.560 | 0.663 | 0.540 | 0.757 | 0.578 | 0.726 | 0.579 | 0.602 | 0.530 | 0.868 | 0.625 | 0.576 | 0.511 | 0.565 | 0.513 | 0.865 | 0.621 |
| ETTh2 → ETTm2 | 0.292 | 0.350 | 0.327 | 0.363 | 0.339 | 0.371 | 0.335 | 0.370 | 0.351 | 0.388 | 0.350 | 0.385 | 0.335 | 0.382 | 0.329 | 0.371 | 0.325 | 0.365 | 0.342 | 0.376 |
| ETTm1 → ETTh2 | 0.389 | 0.417 | 0.442 | 0.439 | 0.440 | 0.449 | 0.433 | 0.439 | 0.484 | 0.466 | 0.530 | 0.489 | 0.455 | 0.458 | 0.428 | 0.431 | 0.439 | 0.438 | 0.457 | 0.454 |
| ETTm1 → ETTm2 | 0.268 | 0.323 | 0.304 | 0.347 | 0.311 | 0.343 | 0.313 | 0.348 | 0.303 | 0.340 | 0.310 | 0.347 | 0.319 | 0.363 | 0.297 | 0.330 | 0.296 | 0.334 | 0.322 | 0.354 |
| ETTm2 → ETTh2 | 0.372 | 0.408 | 0.406 | 0.429 | 0.429 | 0.448 | 0.435 | 0.443 | 0.414 | 0.429 | 0.413 | 0.428 | 0.432 | 0.447 | 0.417 | 0.428 | 0.409 | 0.425 | 0.435 | 0.443 |
| ETTm2 → ETTm1 | 0.496 | 0.462 | 0.622 | 0.532 | 0.588 | 0.503 | 0.769 | 0.567 | 0.556 | 0.481 | 0.710 | 0.554 | 0.706 | 0.572 | 0.527 | 0.474 | 0.568 | 0.492 | 0.769 | 0.567 |
| Average | 0.375 | 0.401 | 0.437 | 0.431 | 0.434 | 0.429 | 0.472 | 0.441 | 0.441 | 0.430 | 0.454 | 0.438 | 0.492 | 0.462 | 0.409 | 0.413 | 0.412 | 0.417 | 0.492 | 0.451 |

## 4.3 Few-Shot Learning

**Setups.** Given the remarkable few-shot learning capabilities of LLMs [44], we follow the setup in [22] to assess the few-shot performance of our method. We train the model using limited data (only the first 5% or 10% data of train set) and evaluate it on the whole test set.

**Results.** The brief 5% and 10% few-shot learning results are illustrated in Table 3. Our S²TS-LLM generally outperforms the baselines on different datasets and settings, with the exception of the ETTm1 dataset. A possible reason is that ETTm1 has finer temporal resolution and more complex variations than other datasets. This increases the difficulty for our method to capture sparse spectral patterns from limited data, leading to suboptimal results. However, our S²TS-LLM realizes average MSE reductions of 4.1%, 6.7%, and 8.2% over S2IP-LLM, Time-LLM, and GPT4TS, respectively, under the 5% few-shot setting, and reductions of 5.2%, 17.0%, and 4.8% under the 10% few-shot setting. Compared to conventional models such as RLinear and iTransformer, our S²TS-LLM leverages the generalizability of LLMs and yields relative improvements from 6.4% to 34.8% over two settings. These results highlight the effectiveness of our approach in few-shot learning scenarios.

## 4.4 Zero-Shot Learning

**Setups.** Beyond few-shot learning, prior work [8] has shown that LLMs also hold promise as effective zero-shot forecasters. To explore this potential, we investigate the zero-shot capabilities of our S²TS-LLM under a cross-domain adaptation setting, where the model is first trained on a source dataset and then directly applied to an unseen target dataset.

**Results.** The zero-shot learning results in Table 4 show that our S²TS-LLM remarkably excels over all baselines with 8.3% improvement over second-best RLinear and 13.5% improvement over LLM-based Time-LLM in terms of MSE metric. Experiments in both few-shot and zero-shot settings highlight S²TS-LLM's exceptional performance under data-scarce conditions. We attribute this to our approach's ability to transfer the LLM's linguistic knowledge to time-series reasoning by learning robust priors of temporal and contextual patterns.

## 4.5 Time Series Classification

**Setups.** To assess the model's ability in representation learning, we conduct experiments on sequence-level classification using the setup outlined by [13]. We select 10 multivariate datasets from the UEA Time Series Classification Archive [45], covering the spectral recognition, medical diagnosis, and other tasks. All datasets are preprocessed by following [46], where different subsets have different sequence lengths.

**Results.** As shown in Figure 5, our $S^2$TS-LLM achieves the best performance with an average accuracy of **76.4%**, surpassing the LLM-based GPT4TS (74.0%), CNN-based TimesNet (73.6%), and classical method Rocket (72.5%). It is worth noting that some conventional models achieve competitive performance on forecasting tasks with LLM-based methods, but fail in classification tasks, such as PatchTST (67.4%) and RLinear (67.2%). These observations show that the prior linguistic knowledge of LLMs can help the time series representation learning.

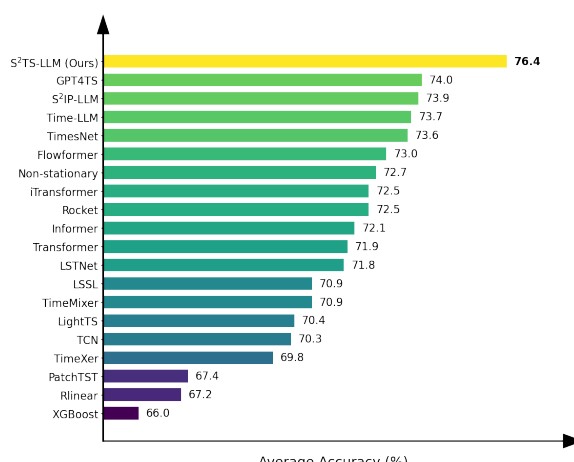

Figure 5: Classification results. The results are averaged from ten datasets of UEA. Detailed results are provided in Appendix E.5.

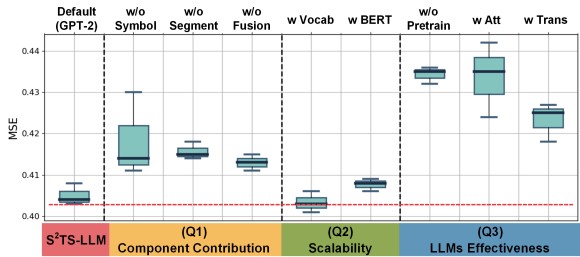

Figure 6: Ablation Study conducted with three experimental repeats. The red dashed line represents the best result of $S^2$TS-LLM. Detailed results are provided in Appendix E.6.

## 4.6 Ablation Study

As depicted in Figure 6, we conduct an ablation study with 8 model variants on ETTh1 to evaluate the (Q1) components' contribution to the framework, (Q2) scalability of the framework, and (Q3) LLM effectiveness to time series tasks. To evaluate Q1, the variants **"w/o Symbol"** and **"w/o Segment"** remove the spectral symbolization and contextual segmentation modules, respectively, and **"w/o Fusion"** replaces our fusion strategy with a simple addition of the LLM outputs. These three variants impair the knowledge transfer from language to time series and lead to performance drops, confirming the importance of each designed module. Then, we evaluate Q2 from two aspects: first, **"w Vocab"** replaces ASCII symbols with vocabulary for textual abstracting, but yields similar performance, verifying the scalability of our symbol-based strategy—it effectively captures spectral patterns while avoiding the need for task-specific vocabulary design; second, **"w BERT"** replaces GPT-2 with BERT [47] and still achieves competitive results, demonstrating the framework's scalability across LLM architectures. Regarding Q3, **"w/o Pretrain"** (random LLM initialization), **"w Att"**, and **"w Trans"** (replace the LLM with a single Attention or Transformer layer) all underperform compared to $S^2$TS-LLM, highlighting the value of pre-trained LLMs in time series analysis.

## 4.7 Visual Interpretation

**Interpretation of Spectral Symbolization.** We provide a case study on ETTh1 to show the effectiveness of spectral symbolization in Figure 7. The top 4 subplots depict the progressive refinement of spectral symbolization, as the generated textual abstractions (green dots) transition from **(a)** a scattered, random distribution to **(d)** a compact distribution that fits the time series space (blue dots). Figure 7 **(e)** visualizes the cosine similarity between textual abstractions and time series in a heatmap form. We observe that most cosine similarity values are close to 1 (indicated by light green), suggesting that the spectral symbolization effectively generates textual abstractions to capture temporal patterns.

**Interpretation of Contextual Segmentation.** Figure 8 presents a case study on ETTh2 to illustrate how contextual segmentation divides a temporal sequence into different blocks. We observe that a time series composed of $Q$ patches is generally divided into three main parts: (1) the initial patches are grouped together as the first segment, as they are farthest from the prediction sequence and capture long-range dependencies; (2) the middle patches are segmented based on their varying trends and patterns; and (3) the final patches, which are closest to the prediction sequence, are separated into a distinct segment to model short-range dependencies.

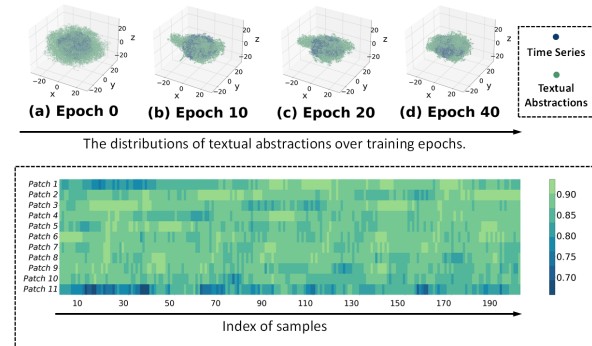

The distributions of textual abstractions over training epochs.

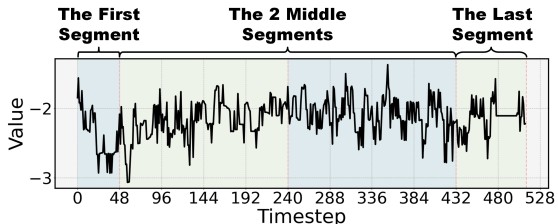

**(e) Similarity between Textual Abstractions and Time Series**

Figure 7: A showcase of spectral symbolization.

## 5 Conclusion

Considering the significant importance of LLMs in complex real-world time series analysis, we empower LLMs' ability to perceive temporal variations and contextual structures. Technically, we revisit the alignment between natural language and time series representations by operating in the frequency domain rather than the time domain, allowing the use of compact symbolic forms to capture frequency semantics.

Figure 8: A showcase of contextual segmentation, where different segments distinguished by blue and green colors can be grouped into three main parts.

Furthermore, we mitigate contextual discrepancies between time series and language by adapting the LLM's positional indices to reflect segment-level temporal structures. Experimental results show that the proposed S$^2$TS-LLM effectively bridges time series and linguistics, enabling LLMs to generalize across diverse time-series tasks while achieving promising performance.

## 6 Acknowledgment

This work is supported in part by the National Key-Research and Development Program of China (Grant No. 2020YFB2104003), the Natural Science Foundation of China (Grant Nos. 62402168, U23A20322), and the Natural Science Foundation of Hunan Province (Grant No. 2024JJ6156).

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

# A Notations

The main notations used in this paper are described in Table 5.

Table 5: A summary of notations.

| Symbol | Description |
|---|---|
| $X, \hat{X}$ | Raw and patched time series |
| $Y, \hat{Y}$ | Ground truth and prediction |
| $T$ | Textual Abstractions of time series |
| $P$ | Positional index of the LLMs |
| $f$ | frequency |
| $v$ | textual token |
| $c$ | Hidden embeddings of GRU |
| $s$ | segment score |
| $E$ | Embeddings of time series or textual abstractions |
| $H$ | Task-specific prompts |
| $\tau$ | Learnable timestamp parameter |
| $Z$ | Inputs of the LLMs |
| $O$ | Outputs of the LLMs |
| $A$ | Cosine similarity between time series and textual abastractions |
| $L$ | Length of input time series |
| $N$ | Number of variants |
| $Q$ | Number of patches |
| $M$ | Patch size |
| $R$ | Stride |
| $B$ | Number of segments |

# B Implementation

## B.1 Dataset Descriptions

We conduct experiments on 23 real-world datasets to evaluate the performance of our proposed $S^2$TS-LLM. These datasets span domains such as energy, weather, transportation, finance, and healthcare, and have been widely used as benchmarks for time series analysis tasks, including forecasting and classification. Dataset statistics are summarized in Table 6.

Regarding the long-term forecasting task, we utilize 7 well-acknowledged multivariate datasets, including: (1) **Weather** comprises 21 meteorological variables collected at 10-minute intervals by the Weather Station of the Max Planck Biogeochemistry Institute in 2020. (2) **Electricity (ECL)** provides hourly measurements of electricity consumption from 321 consumers. (3) **Traffic** contains hourly road occupancy rates collected from 862 sensors installed along the San Francisco Bay Area freeways. (4) **Electricity Transformer Temperature (ETT)** is a key indicator reflecting long-term electric power usage. The dataset provides six power load features and one target variable, the 'oil temperature', collected over two years from two countries in China. It is divided into 4 distinct subsets where ETTh1 and ETTh2 contain hourly recordings, and ETTm1 and ETTm2 are sampled every 15 minutes. We note that the ETTh1, ETTh2, ETTm1, and ETTm2 datasets are additionally employed for few-shot and zero-shot learning.

Regarding the short-term forecasting task, we adopt the M4 benchmark for performance evaluation. The M4 dataset consists of 100,000 time series from diverse domains, including business, finance, and economics. These time series data are grouped into six distinct subsets with sampling rates ranging from yearly to hourly.

Regarding the time series classification task, we conduct experiments on the UEA Multivariate Time Series Classification Archive [45], which comprises 30 sub-datasets of time series gathered from diverse fields. Following [13], we select 10 multivariate subsets from the UEA archive for classification tasks, covering gesture, action, and audio recognition, medical diagnosis, and other practical tasks.

Table 6: Dataset specifications across different tasks. Notably, the few-shot and zero-shot learning tasks utilize the same datasets as long-term forecasting, but only sample the first 5% or 10% of the data for training.

| Tasks | Dataset | Dim | Series Length | Dataset Size | Information (Frequency) |
|---|---|---|---|---|---|
| Long-term Forecasting | ETTm1, ETTm2 | 7 | {96, 192, 336, 720} | (34465, 11521, 11521) | Electricity (15 mins) |
| | ETTh1, ETTh2 | 7 | {96, 192, 336, 720} | (8545, 2881, 2881) | Electricity (Hourly) |
| | Electricity | 321 | {96, 192, 336, 720} | (18317, 2633, 5261) | Electricity (Hourly) |
| | Traffic | 862 | {96, 192, 336, 720} | (12185, 1757, 3509) | Transportation (Hourly) |
| | Weather | 21 | {96, 192, 336, 720} | (36792, 5271, 10540) | Weather (10 mins) |
| Short-term Forecasting | M4-Yearly | 1 | 6 | (23000, 0, 23000) | Demographic |
| | M4-Quarterly | 1 | 8 | (24000, 0, 24000) | Finance |
| | M4-Monthly | 1 | 18 | (48000, 0, 48000) | Industry |
| | M4-Weekly | 1 | 13 | (359, 0, 359) | Macro |
| | M4-Daily | 1 | 14 | (4227, 0, 4227) | Micro |
| | M4-Hourly | 1 | 48 | (414, 0, 414) | Other |
| Classification (UEA) | EthanolConcentration | 3 | 1751 | (261, 0, 263) | Alcohol Industry |
| | FaceDetection | 144 | 62 | (5890, 0, 3524) | Face (250Hz) |
| | Handwriting | 3 | 152 | (150, 0, 850) | Handwriting |
| | Heartbeat | 61 | 405 | (204, 0, 205) | Heart Beat |
| | JapaneseVowels | 12 | 29 | (270, 0, 370) | Voice |
| | PEMS-SF | 963 | 144 | (267, 0, 173) | Transportation (Daily) |
| | SelfRegulationSCP1 | 6 | 896 | (268, 0, 293) | Health (256Hz) |
| | SelfRegulationSCP2 | 7 | 1152 | (200, 0, 180) | Health (256Hz) |
| | SpokenArabicDigits | 13 | 93 | (6599, 0, 2199) | Voice (11025Hz) |
| | UWaveGestureLibrary | 3 | 315 | (120, 0, 320) | Gesture |

## B.2 Baselines

We compare our $S^2$TS-LLM against a series of well-known and advanced baselines for forecasting tasks, including LLM-based models ($S^2$IP-LLM [7], Time-LLM [9], GPT4TS [22]), Transformer-based models (TimeXer [38], iTransformer [39], PatchTST [40]), MLP-based models (TimeMixer [41], RLinear [17]), and a CNN-based model (TimesNet [13]). The details of the baselines are shown below.

- $S^2$IP-LLM [7]: $S^2$IP-LLM activates the LLMs for time series tasks by aligning the pre-trained semantic space with time series representation space. Technically, it encodes the large and dense word token embeddings into a small set of semantic anchors and then retrieves the top-$k$ anchors that best match the decomposed time series embeddings, using them as prompts to bootstrap the LLMs' understanding of time series.

- Time-LLM [9]: Time-LLM introduces a reprogramming idea that aligns time series data with textual prototypes. Along with prefix prompts that provide descriptions and statistics of specific datasets, the reprogramming framework achieves cross-modal adaptation of LLMs for time series forecasting without any fine-tuning. For a fair comparison, this paper adopts GPT-2 as the backbone LLM for Time-LLM rather than its default setup (Llama-7B).

- GPT4TS [22]: GPT4TS leverages pre-trained LLMs for general time series analysis through a simple yet effective architecture called Frozen Pre-trained Transformer (FPT). FPT freezes the self-attention and feedforward layers while fine-tuning the layer normalization and positional embeddings, achieving promising performance across various time series tasks.

- TimeMixer [41]: TimeMixer proposes an MLP-based architecture designed to capture the multiscale temporal variations. Specifically, it decomposes and mixes the multiscale time series in both fine-to-coarse and coarse-to-fine directions, enabling comprehensive aggregation of microscopic and macroscopic information for time series learning.

- TimeXer [38]: TimeXer leverages the patch-level and variate-level representation ability, as well as the attention mechanism in Transformer, to model interactive dependencies between endogenous and exogenous variables, effectively capturing the causal relationships inherent in multivariate time series for forecasting.

- iTransformer [39]: iTransformer repurposes the canonical Transformer to focus on multivariate dependency modeling by inverting the embedding dimensions in the attention and feed-forward layers, achieving strong performance on real-world datasets.

- RLinear [17]: RLinear shows that a single linear layer, when combined with reversible normalization and channel independence techniques, can achieve efficient yet competitive performance compared to specialized and complex networks.

- PatchTST [40]: PatchTST introduces the concepts of patching and channel independence by segmenting time series into patches while preserving independence across variates. These designs enable Transformers to effectively capture both local and global dependencies, leading to improved performance.

- TimesNet [13]: TimesNet introduces a general time-series analysis approach based on frequency analysis and convolutional operations. It captures both intra-period and inter-period variations by transforming a 1D time series into a 2D representation.

Besides, a broader set of models is further employed for classification comparison, including Transformer-based models (Non-stationary Transformer [48], FlowFormer [49], Informer [50], Transformer [51]), MLP-based model (LightTS [52]), CNN-based model (TCN [53]), RNN-based models (LSSL [54], LSTNet [2]), and classical models (Rocket [55], XGBoost [56]). The details of the baselines are as follows.

- Non-stationary Transformer [48]: Non-stationary Transformer addresses the non-stationarity problem by employing stationary and de-stationary attention modules to restore and recover non-stationary information in time series, thereby improving the model's ability to distinguish temporal variations for forecasting.

- FlowFormer [49]: FlowFormer adopts a Transformer-based architecture inspired by optical flow estimation, utilizing a 4D cost volume to model fine-grained correspondences, which can be adapted for time series classification tasks.

- Informer [50]: Informer explores the sparsity of the attention mechanism to capture long-term temporal dependencies while reducing time and space complexity by identifying dominant features during attention operations.

- Transformer [51]: Transformer proposes an encoder-decoder architecture based on attention mechanisms to capture global dependencies in sequences, outperforming conventional RNNs/CNNs in speed and accuracy due to parallel processing and global context modeling.

- LightTS [52]: LightTS presents an efficient MLP-based framework that employs two distinct downsampling strategies to extract both short-term and long-term patterns in time series.

- TCN [53]: TCN introduces a deep neural network based on dilated causal convolutions, enabling efficient modeling of long-range dependencies in multivariate time series. In addition, it is trained in an unsupervised manner using a time-based triplet loss, which encourages similar subseries to have similar embeddings.

- LSSL [54]: LSSL introduces a structured state space model that reparameterizes state space models (SSMs) using a structured normal-plus-low-rank (NPLR) decomposition, enabling recurrent networks to effectively capture long-range dependencies in time series.

- LSTNet [2]: LSTNet is an autoregressive framework that combines a convolutional layer with a recurrent layer to capture short-term local dependencies and long-term periodic patterns with varying scales, enhancing the robustness of time series analysis.

- Rocket [55]: ROCKET is a classical time series classification method that transforms input sequences using thousands of randomly generated convolutional kernels, achieving state-of-the-art accuracy with significantly lower computational cost.

- XGBoost [56]: XGBoost builds upon traditional gradient tree boosting by improving sparsity handling and large-scale scalability, resulting in an efficient and flexible system applicable to a wide range of machine learning tasks, including time series analysis.

### B.3   Implementation Details

All deep neural networks are implemented in PyTorch and trained on a Linux server (CPU: Intel(R) Xeon(R) Platinum 8452Y @ 2.00GHz, GPU: NVIDIA H800 GPU 80GB). To ensure fair comparisons, we adhere to the experimental configurations outlined in [13] to maintain a uniform evaluation procedure across all baselines. For baselines, we strictly follow their provided experimental settings or cite their performance if applicable [27, 41, 38].

Most of the training settings of our proposed framework are based on [22]. Specifically, we implement the proposed $S^2$TS-LLM using the first 4 layers of GPT-2 as the backbone LLM. Regarding the forecasting task, we train our $S^2$TS-LLM using the Adam optimizer with a learning rate of 0.0001, a decay rate $\beta = (0.9, 0.999)$, and a batch size of 256. A cosine annealing schedule with $T_{max} = 20$ and $\eta_{min} = 10^{-8}$ is used to adjust the model optimization. Early stopping is configured throughout the training process with a patience of 10. MSE loss is adopted for long-term forecasting, while SMAPE loss is used for short-term predictions. The top-$k$ frequency in the spectral symbolization module is set to 5, and the segment score threshold $\gamma$ in the contextual segmentation module is set to 0.1. The input time series is split into $G = 2$ slices for task prompt augmentation. Regarding the classification task, we train our $S^2$TS-LLM using the RAdam optimizer with a learning rate selected from $\{0.0005, 0.005\}$, and a cross-entropy loss is used for model optimization.

Regarding the metrics, we adopt the Mean Square Error (MSE) and Mean Absolute Error (MAE) for long-term forecasting. For the short-term forecasting, following [9], we adopt the Symmetric Mean Absolute Percentage Error (SMAPE), Mean Absolute Scaled Error (MASE), and Overall Weighted Average (OWA) as the metrics, where OWA is a special metric used in M4 competition. These metrics are defined as follows:

$$\text{MSE} = \frac{1}{H} \sum_{h=1}^{T} (\boldsymbol{Y}_l - \hat{\boldsymbol{Y}}_l)^2, \qquad\qquad \text{MAE} = \frac{1}{L_p} \sum_{l=1}^{L_p} |\boldsymbol{Y}_l - \hat{\boldsymbol{Y}}_l|,$$

$$\text{SMAPE} = \frac{200}{L_p} \sum_{l=1}^{L_p} \frac{|\boldsymbol{Y}_l - \hat{\boldsymbol{Y}}_l|}{|\boldsymbol{Y}_l| + |\hat{\boldsymbol{Y}}_l|}, \qquad\qquad \text{MAPE} = \frac{1}{L_p} \sum_{l=1}^{L_p} \frac{|\boldsymbol{Y}_l - \hat{\boldsymbol{Y}}_l|}{|\boldsymbol{Y}_l|},$$

$$\text{MASE} = \frac{1}{L_p} \sum_{l=1}^{L_p} \frac{|\boldsymbol{Y}_l - \hat{\boldsymbol{Y}}_l|}{\frac{1}{L_p - s} \sum_{j=s+1}^{L_p} |\boldsymbol{Y}_j - \boldsymbol{Y}_{j-s}|}, \qquad \text{OWA} = \frac{1}{2} \left[ \frac{\text{SMAPE}}{\text{SMAPE}_{\text{Naïve2}}} + \frac{\text{MASE}}{\text{MASE}_{\text{Naïve2}}} \right].$$

## C   More Experiments

### C.1   Analysis of Frequency Learning

Since the proposed method extracts spectral information to bootstrap LLMs' time-series comprehension, we further evaluate its frequency-domain learning capacity by examining magnitude sensitivity and trend preservation capability. Regarding magnitude sensitivity, we compute cosine similarity and Pearson correlation between the frequency components of predictions and ground truth. For trend preservation, we measure the dominant frequency overlap to assess how well each model captures underlying trends. Table 7 presents a comparative analysis of our $S^2$TS-LLM against two representative frequency-based models (FEDformer [18] and FiLM [16]), where our $S^2$TS-LLM consistently outperforms both baselines, demonstrating its effectiveness in preserving frequency-domain contributions for time-series modeling.

Table 7: Comparison of different methods in terms of magnitude sensitivity and trend preservation capability.

| Dataset | Frequency Analysis | | S$^2$TS-LLM (Ours) | FEDformer | FiLM |
|---------|------|------|------|------|------|
| ETTh1 | Magnitude Test | Cosine Similarity ↑ | 0.926 | 0.919 | 0.912 |
| | | Pearson Correlation ↑ | 0.878 | 0.865 | 0.857 |
| | Trend Test | Dominant Frequency Overlap ↑ | 0.523 | 0.503 | 0.507 |
| ETTh2 | Magnitude Test | Cosine Similarity ↑ | 0.987 | 0.985 | 0.982 |
| | | Pearson Correlation ↑ | 0.985 | 0.982 | 0.979 |
| | Trend Test | Dominant Frequency Overlap ↑ | 0.851 | 0.844 | 0.846 |

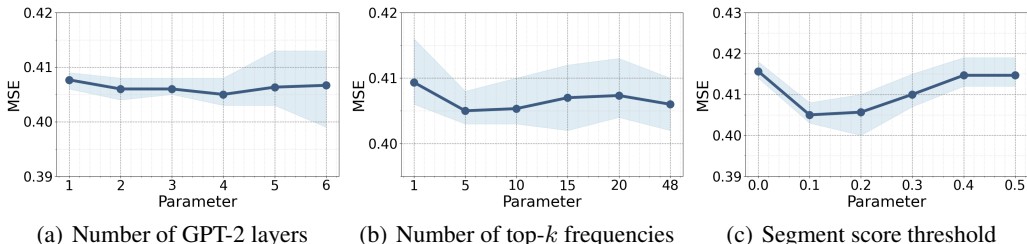

(a) Number of GPT-2 layers    (b) Number of top-$k$ frequencies    (c) Segment score threshold

Figure 9: Sensitivity analysis of hyperparameters (a) number of GPT-2 layers, (b) number of top-$k$ frequencies, and (c) segment score threshold on the ETTh1 dataset.

## C.2 Hyperparameter Sensitivity

We perform a hyperparameter sensitivity analysis on the three key hyperparameters of the proposed S$^2$TS-LLM: the number of GPT-2 layers, the number of top-$k$ frequencies, and the segment score threshold $\gamma$. This analysis is conducted on the ETTh1 dataset by averaging results across four prediction horizons $\{96, 192, 336, 720\}$ under different hyperparameter settings. Figure 9 presents the results averaged over three independent runs for each hyperparameter, where the dark blue lines represent the average results from three runs and the light blue boundaries indicate the error bars.

- **Number of GPT-2 layers.** GPT-2 typically consists of 12 transformer layers. We assess the impact of GPT-2 network depth by varying the number of layers from 1 to 6. As shown in Figure 9(a), our S$^2$TS-LLM exhibits low sensitivity to the number of layers. This can be attributed to the use of minimalist symbolic abstractions, which effectively align time series with natural language and reduce the reliance on deep LLM layers for multimodal understanding. While increasing the number of layers can reduce prediction error, deeper models may introduce the overfitting problem, leading to unstable results—as evidenced by the 6-layer variant, which shows the lowest MSE but the highest variance. To trade off performance and computational efficiency, we adopt the first 4 layers of GPT-2 as the backbone LLM in our framework.

- **Number of top-$k$ frequencies.** Due to the symmetry of the Fast Fourier Transform, the number of top-$k$ frequencies is selected from the candidate set $\{1, 5, 10, 15, 20, 24\}$, ensuring it does not exceed half the feature dimension of the patched input $\hat{X}$ (48 dimensions, as used in [22]). As depicted in Figure 9 (b), neither too few nor too many top-$k$ frequencies lead to satisfactory results: a smaller $k$ fails to capture sufficient frequency patterns, while a larger $k$ introduces noisy spectral components. Based on this analysis, we set $k = 5$ to achieve the best performance.

- **Segment score threshold.** We select the segment score threshold $\gamma$ from the range $[0.0, 0.5]$, and the sensitivity results are illustrated in Figure 9 (c). When $\gamma = 0$, each input patch is treated as an individual segment, disabling the contextual segmentation mechanism and resulting in the worst performance. As $\gamma$ increases to 0.5, the segmentation tends to form overly long segments or even a single segment for the entire input, which diminishes the structural granularity of the time series and leads to performance degradation. Based on these observations, we set $\gamma = 0.1$ to achieve optimal performance.

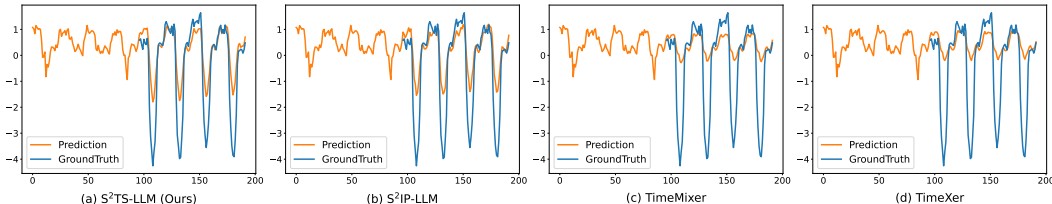

Figure 10: Long-term forecasting cases from ETTh1 by different models under the input-96-predict-96 settings. Blue lines are the ground truths and orange lines are the model predictions.

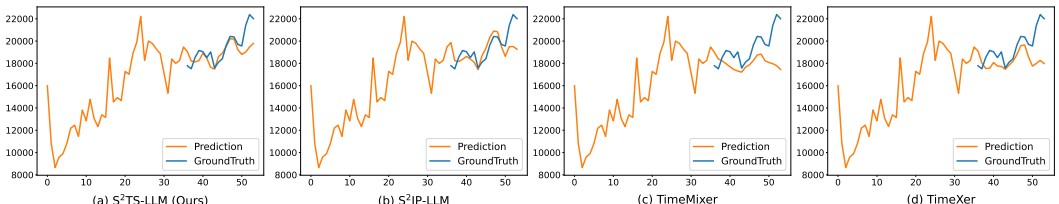

Figure 11: Short-term forecasting cases from the M4 dataset by different models under the input-36-predict-18 settings. Blue lines are the ground truths and orange lines are the model predictions.

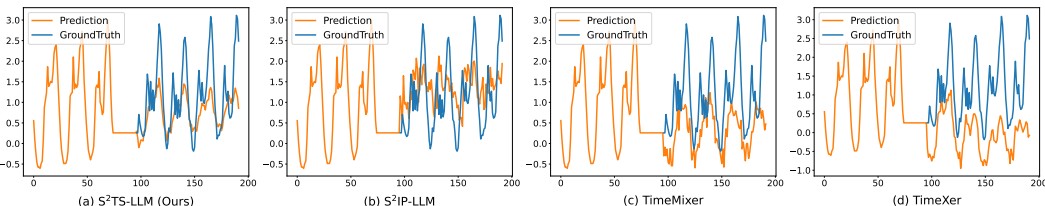

Figure 12: Few-shot forecasting cases from ETTh1 by different models under the input-96-predict-96 settings. Blue lines are the ground truths and orange lines are the model predictions.

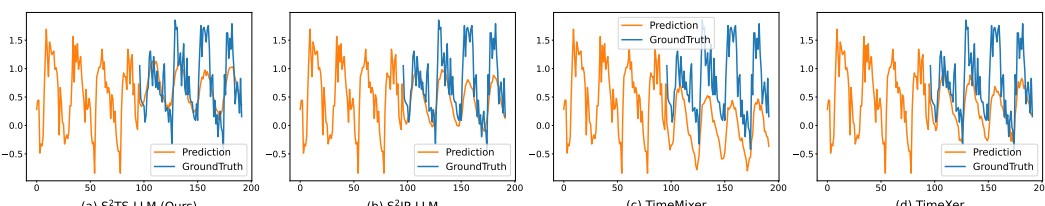

Figure 13: Zero-shot forecasting cases from ETTh2→ETTh1 by different models under the input-96-predict-96 settings. Blue lines are the ground truths and orange lines are the model predictions.

### C.3 Forecasting Visualization

To evaluate the forecasting capabilities of the proposed $S^2$TS-LLM, Figures 10 to 13 present visualization examples across long-term forecasting, short-term forecasting, few-shot learning, and zero-shot learning tasks, which are compared against baseline models, including $S^2$IP-LLM [7], TimeMixer [41], and TimeXer [38]. We observe that the datasets exhibit complex variations that pose significant challenges for prediction. For instance, the ETTh1 example in Figure 10 begins with regular periodic fluctuations, followed by a sequence of abrupt drops and spikes. Despite such complexity, the visualizations demonstrate that $S^2$TS-LLM achieves good predictive performance across various tasks. This is especially evident in few-shot and zero-shot scenarios, where $S^2$TS-LLM successfully captures future trends, whereas the baseline models often struggle.

Table 8: Training parameters and Training/Inference Cost Comparison

| Type | Model | Training Params | Training Params Percentage (%) | Training Time per Step (s) | Inference Time per Batch (s) |
|---|---|---|---|---|---|
| Conventional Specialized Model | TimesNet-32 | 4.77M | 100 | 0.207 | 0.156 |
| | TimesNet-768 | 107.63M | 100 | 1.020 | 0.856 |
| LLM-based Model | GPT4TS(4)-768 | 1.57M | 2 | 0.029 | 0.031 |
| | Time-LLM(4)-768 | 54.25M | 46 | 0.551 | 0.941 |
| | S$^2$IP-LLM(4)-768 | 49.15M | 44 | 0.562 | 1.202 |
| | **S$^2$TS-LLM(4)-768 (Ours)** | 2.71M | 4 | 0.158 | 0.189 |

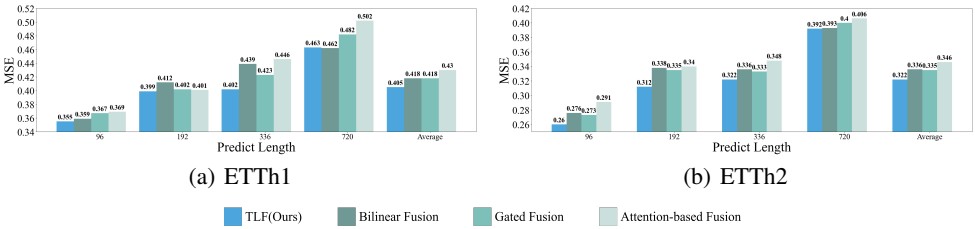

(a) ETTh1                          (b) ETTh2

Figure 14: Comparison of different fusion strategy on the ETTh1 and ETTh2 datasets, where lower MSE values indicate better performance.

## C.4 Computational Cost

The computational cost is a critical factor in assessing the practicality of LLM-based models. To evaluate this, we compare our S$^2$TS-LLM with a specialized model (TimesNet) and three LLM-based models (GPT4TS, Time-LLM, and S$^2$IP-LLM) in terms of training and inference costs. Specifically, TimesNet is tested in two setups with embedding dimensions of 32 and 768, the latter matching the embedding size of GPT-2. Regarding the LLM-based baselines, the default settings of LLM-based S$^2$IP-LLM, Time-LLM, and GPT4TS utilize the first 6, 12, and 6 hidden layers of GPT-2, respectively. To ensure a fair comparison, we uniformly configure all LLM-based baselines with 4 layers, consistent with the architecture of our proposed model. The computational cost analysis is conducted on the ETTh1 dataset with a batch size of 128, using an 80G H800 GPU.

As shown in Table 8, our S$^2$TS-LLM achieves competitive efficiency and parameter count compared to baseline methods. Particularly, our S$^2$TS-LLM outperforms TimesNet in training efficiency while delivering superior forecasting accuracy, underscoring the practicality of our approach. Compared to LLM-based methods, S$^2$TS-LLM incurs a slightly higher computational cost than GPT4TS, primarily due to the additional overhead introduced by our cross-modal alignment and prompt augmentation. Importantly, while sharing the same objective of cross-modal alignment with S$^2$IP-LLM and Time-LLM, our method achieves superior computational and memory efficiency. This highlights that the sparsity inherent in frequency-domain signals can significantly reduce the complexity of aligning time series with natural language, validating the efficacy of our spectral symbolization paradigm.

## C.5 Effectiveness of Time-Linguistic Fusion

Inspire by the RNNs, our Time-Linguistic Fusion (TLF) adopts a gated residual update mechanism $(\boldsymbol{O}^{(n)} = \boldsymbol{O}_X^{(n)} + \boldsymbol{A}^{(n)} \odot \sigma(\boldsymbol{O}_T^{(n)}) \odot \boldsymbol{O}_X^{(n)})$ to fuse textual and time-series outputs. To assess the effectiveness of this design, we compare it against several alternative fusion strategies, including bilinear fusion $(z = x^T W y)$, gated fusion $(z = \alpha x + (1 - \alpha)y)$, and attention-based fusion (Softmax $\left( \frac{(yW_Q)(xW_K)^T}{\sqrt{d}} (xW_V) \right)$), where $x = O_X^{(n)}$ and $y = O_T^{(n)}$. MSE results in Figure 14 demonstrate that our proposed TLF consistently outperforms these alternatives. This superior performance arises because the sigmoid function $\sigma(\cdot)$ softly modulates frequency information by smoothly mapping spectral features into a bounded embedding space $(0, 1)$, rather than discarding them. Furthermore, the residual fusion strategy is particularly well-suited for integrating auxiliary frequency cues with raw numerical features: frequency-domain signals guide LLMs semantically

Table 9: Performance comparison of different methods across varying lookback lengths.

| Dataset | Lookback Length | 512 | | 336 | | 192 | | 96 | | 48 | |
| | Metric | MSE↓ | MAE↓ | MSE↓ | MAE↓ | MSE↓ | MAE↓ | MSE↓ | MAE↓ | MSE↓ | MAE↓ |
|---|---|---|---|---|---|---|---|---|---|---|---|
| ETTh1 | $S^2$TS-LLM (Ours) | 0.355 | 0.387 | 0.365 | 0.389 | 0.372 | 0.391 | 0.378 | 0.391 | 0.398 | 0.411 |
| | $S^2$IP-LLM | 0.368 | 0.403 | 0.367 | 0.398 | 0.368 | 0.393 | 0.379 | 0.394 | 0.390 | 0.401 |
| | GPT4TS | 0.376 | 0.397 | 0.378 | 0.399 | 0.380 | 0.401 | 0.383 | 0.401 | 0.385 | 0.403 |
| ETTh2 | $S^2$TS-LLM (Ours) | 0.260 | 0.329 | 0.281 | 0.335 | 0.288 | 0.338 | 0.291 | 0.336 | 0.310 | 0.350 |
| | $S^2$IP-LLM | 0.284 | 0.345 | 0.282 | 0.343 | 0.288 | 0.343 | 0.293 | 0.343 | 0.301 | 0.344 |
| | GPT4TS | 0.285 | 0.342 | 0.286 | 0.345 | 0.294 | 0.349 | 0.295 | 0.349 | 0.299 | 0.350 |

rather than numerically, and residual pathways enable smoother integration without disrupting the structure of the original signal.

## C.6   Analysis of Performance Gain

To investigate when our $S^2$STS-LLM works well and when it fails, we follow prior work [9, 19] to evaluate model performance under different lookback lengths $\{512, 336, 192, 96, 48\}$. The detailed MSE results on the ETTh1 and ETTh2 datasets are shown in Table 9. We observe that $S^2$STS-LLM outperforms the other two LLM-based methods, $S^2$IP-LLM and GPT4TS, when the lookback length is between 512 and 96. However, as the lookback length decreases to 48, $S^2$STS-LLM experiences a significant performance drop and underperforms compared to the $S^2$IP-LLM and GPT4TS. This is consistent with Table 2's observation that $S^2$STS-LLM achieves suboptimal results on short-term forecasting tasks. The main reason is that $S^2$STS-LLM uses FFT to extract dominant frequency components, which leads to further information compression when the lookback window is short. As a result, the generated compact textual descriptions may not provide LLMs with sufficient information to bridge the semantic gap between linguistics and time series. Therefore, our method is better suited for long-sequence scenarios. For example, our model shows significant gains in both long-term forecasting and zero-shot learning tasks (refer to Tables 1 and 4), where it can capture enough dominant frequency components to distinguish different temporal patterns.

## D   Discussion

**Impact Statement.** This work presents a step forward in time series forecasting by exploring how time series data can be aligned with the representational space and contextual structures of the LLMs. Our proposed frequency-based alignment approach offers a compact and efficient alternative for cross-modal adaptation, highlighting the potential of leveraging the frequency domain rather than the time domain for time-series-driven LLMs. Beyond representation, our study also underscores the value of contextual alignment in adapting LLMs to time series tasks. We believe that as the LLMs develop a deeper understanding of time series, conventional time series analysis will gradually advance toward a more universal form of artificial intelligence. In particular, the extensive internal knowledge and reasoning capabilities of LLMs hold great promise for addressing complex time series tasks and improving decision-making across critical domains such as finance, healthcare, and environmental monitoring.

**Limitation and Future Work.** $S^2$TS-LLM demonstrates favorable performance and reasonable efficiency across various time series analysis tasks, including forecasting and classification. However, we observe that the use of time–frequency transformation may intensify to information compression when the input sequence is relatively short, resulting in performance degradation (refer to Table 2 and Table 9). To address this limitation, future work will explore alternative frequency representations to augment frequency information, such as multi-scale fusion of frequency features or contrastive learning across different frequency views. Moreover, this study focuses solely on cross-modal alignment between time series and linguistics, neglecting the visual modality of time series. Incorporating time series, computer vision, and natural language into a unified multi-modal framework is a promising direction. Lastly, given its application-oriented design, we plan to extend $S^2$TS-LLM to broader tasks such as anomaly detection, imputation, and spatio-temporal analysis, while further improving its computational efficiency to enhance practical value.

Table 10: Full long-term forecasting results. The prediction horizons are set to $\{96, 192, 336, 720\}$. A lower value indicates better performance. Red denotes the best performance, while Blue indicates the second-best.

| Methods | | S$^2$TS-LLM | | S$^2$ IP-LLM | | Time-LLM | | GPT4TS | | TimeMixer | | TimeXer | | iTransformer | | RLinear | | PatchTST | | TimesNet | |
|---|---|---|---|---|---|---|---|---|---|---|---|---|---|---|---|---|---|---|---|---|---|
| Metric | | MSE | MAE | MSE | MAE | MSE | MAE | MSE | MAE | MSE | MAE | MSE | MAE | MSE | MAE | MSE | MAE | MSE | MAE | MSE | MAE |
| Weather | 96 | 0.148 | 0.200 | 0.149 | 0.200 | 0.163 | 0.210 | 0.162 | 0.212 | 0.163 | 0.209 | 0.157 | 0.205 | 0.174 | 0.214 | 0.192 | 0.232 | 0.177 | 0.218 | 0.172 | 0.220 |
| | 192 | 0.191 | 0.241 | 0.195 | 0.244 | 0.205 | 0.245 | 0.204 | 0.248 | 0.208 | 0.250 | 0.204 | 0.247 | 0.221 | 0.254 | 0.240 | 0.271 | 0.225 | 0.259 | 0.219 | 0.261 |
| | 336 | 0.244 | 0.283 | 0.246 | 0.280 | 0.257 | 0.287 | 0.254 | 0.286 | 0.251 | 0.287 | 0.261 | 0.290 | 0.278 | 0.296 | 0.292 | 0.307 | 0.278 | 0.297 | 0.280 | 0.306 |
| | 720 | 0.319 | 0.338 | 0.320 | 0.336 | 0.323 | 0.332 | 0.326 | 0.337 | 0.339 | 0.341 | 0.340 | 0.341 | 0.358 | 0.349 | 0.364 | 0.353 | 0.354 | 0.348 | 0.365 | 0.359 |
| | Avg | 0.225 | 0.265 | 0.228 | 0.265 | 0.237 | 0.269 | 0.237 | 0.270 | 0.240 | 0.271 | 0.241 | 0.271 | 0.258 | 0.278 | 0.272 | 0.291 | 0.259 | 0.281 | 0.259 | 0.287 |
| ECL | 96 | 0.133 | 0.228 | 0.138 | 0.234 | 0.140 | 0.236 | 0.139 | 0.238 | 0.153 | 0.247 | 0.140 | 0.242 | 0.148 | 0.240 | 0.201 | 0.281 | 0.195 | 0.285 | 0.168 | 0.272 |
| | 192 | 0.151 | 0.245 | 0.153 | 0.252 | 0.150 | 0.249 | 0.153 | 0.251 | 0.166 | 0.256 | 0.157 | 0.256 | 0.162 | 0.253 | 0.201 | 0.283 | 0.199 | 0.289 | 0.184 | 0.289 |
| | 336 | 0.168 | 0.261 | 0.169 | 0.270 | 0.168 | 0.267 | 0.169 | 0.266 | 0.185 | 0.277 | 0.176 | 0.275 | 0.178 | 0.269 | 0.215 | 0.298 | 0.215 | 0.305 | 0.198 | 0.300 |
| | 720 | 0.205 | 0.291 | 0.204 | 0.293 | 0.209 | 0.302 | 0.206 | 0.297 | 0.225 | 0.310 | 0.211 | 0.306 | 0.225 | 0.317 | 0.257 | 0.331 | 0.256 | 0.337 | 0.220 | 0.320 |
| | Avg | 0.164 | 0.256 | 0.166 | 0.262 | 0.167 | 0.264 | 0.167 | 0.263 | 0.182 | 0.272 | 0.171 | 0.270 | 0.178 | 0.270 | 0.219 | 0.298 | 0.216 | 0.304 | 0.192 | 0.295 |
| Traffic | 96 | 0.364 | 0.263 | 0.379 | 0.274 | 0.384 | 0.278 | 0.388 | 0.282 | 0.462 | 0.285 | 0.428 | 0.271 | 0.395 | 0.268 | 0.649 | 0.389 | 0.462 | 0.295 | 0.593 | 0.321 |
| | 192 | 0.389 | 0.275 | 0.397 | 0.282 | 0.398 | 0.286 | 0.407 | 0.290 | 0.473 | 0.296 | 0.448 | 0.282 | 0.417 | 0.276 | 0.601 | 0.366 | 0.466 | 0.296 | 0.617 | 0.336 |
| | 336 | 0.395 | 0.275 | 0.407 | 0.289 | 0.408 | 0.289 | 0.412 | 0.294 | 0.498 | 0.296 | 0.473 | 0.289 | 0.433 | 0.283 | 0.609 | 0.369 | 0.482 | 0.304 | 0.629 | 0.336 |
| | 720 | 0.436 | 0.297 | 0.440 | 0.301 | 0.436 | 0.303 | 0.450 | 0.312 | 0.506 | 0.313 | 0.516 | 0.302 | 0.467 | 0.302 | 0.647 | 0.387 | 0.514 | 0.322 | 0.640 | 0.350 |
| | Avg | 0.396 | 0.278 | 0.405 | 0.286 | 0.407 | 0.289 | 0.414 | 0.294 | 0.484 | 0.297 | 0.466 | 0.287 | 0.428 | 0.282 | 0.626 | 0.378 | 0.481 | 0.304 | 0.620 | 0.336 |
| ETTh1 | 96 | 0.355 | 0.387 | 0.367 | 0.398 | 0.383 | 0.404 | 0.376 | 0.397 | 0.375 | 0.400 | 0.382 | 0.403 | 0.386 | 0.405 | 0.386 | 0.395 | 0.414 | 0.419 | 0.384 | 0.402 |
| | 192 | 0.399 | 0.419 | 0.402 | 0.422 | 0.427 | 0.431 | 0.416 | 0.418 | 0.429 | 0.421 | 0.429 | 0.435 | 0.441 | 0.436 | 0.437 | 0.424 | 0.460 | 0.445 | 0.436 | 0.429 |
| | 336 | 0.402 | 0.432 | 0.432 | 0.451 | 0.430 | 0.436 | 0.442 | 0.433 | 0.484 | 0.458 | 0.468 | 0.448 | 0.487 | 0.458 | 0.479 | 0.446 | 0.501 | 0.466 | 0.491 | 0.469 |
| | 720 | 0.463 | 0.478 | 0.472 | 0.474 | 0.465 | 0.469 | 0.477 | 0.456 | 0.498 | 0.482 | 0.469 | 0.461 | 0.503 | 0.491 | 0.481 | 0.470 | 0.500 | 0.488 | 0.521 | 0.500 |
| | Avg | 0.404 | 0.429 | 0.418 | 0.436 | 0.426 | 0.435 | 0.427 | 0.426 | 0.447 | 0.440 | 0.437 | 0.437 | 0.454 | 0.447 | 0.446 | 0.434 | 0.469 | 0.454 | 0.458 | 0.450 |
| ETTh2 | 96 | 0.260 | 0.329 | 0.284 | 0.345 | 0.293 | 0.348 | 0.285 | 0.342 | 0.289 | 0.341 | 0.286 | 0.338 | 0.297 | 0.349 | 0.288 | 0.338 | 0.302 | 0.348 | 0.340 | 0.374 |
| | 192 | 0.312 | 0.363 | 0.349 | 0.387 | 0.356 | 0.391 | 0.354 | 0.389 | 0.372 | 0.392 | 0.363 | 0.389 | 0.380 | 0.400 | 0.374 | 0.390 | 0.388 | 0.400 | 0.402 | 0.414 |
| | 336 | 0.322 | 0.380 | 0.368 | 0.417 | 0.372 | 0.408 | 0.373 | 0.407 | 0.386 | 0.414 | 0.414 | 0.423 | 0.428 | 0.432 | 0.415 | 0.426 | 0.426 | 0.433 | 0.452 | 0.452 |
| | 720 | 0.392 | 0.430 | 0.419 | 0.445 | 0.421 | 0.446 | 0.406 | 0.441 | 0.412 | 0.434 | 0.408 | 0.432 | 0.427 | 0.445 | 0.420 | 0.440 | 0.431 | 0.446 | 0.462 | 0.468 |
| | Avg | 0.321 | 0.376 | 0.355 | 0.399 | 0.361 | 0.398 | 0.354 | 0.394 | 0.364 | 0.395 | 0.367 | 0.396 | 0.383 | 0.407 | 0.374 | 0.398 | 0.387 | 0.407 | 0.414 | 0.427 |
| ETTm1 | 96 | 0.282 | 0.343 | 0.291 | 0.348 | 0.294 | 0.345 | 0.292 | 0.346 | 0.318 | 0.356 | 0.334 | 0.368 | | | 0.355 | 0.376 | 0.329 | 0.367 | 0.338 | 0.375 |
| | 192 | 0.322 | 0.368 | 0.323 | 0.368 | 0.330 | 0.368 | 0.332 | 0.372 | 0.361 | 0.381 | 0.362 | 0.383 | 0.387 | 0.391 | 0.391 | 0.392 | 0.367 | 0.385 | 0.374 | 0.387 |
| | 336 | 0.356 | 0.388 | 0.361 | 0.392 | 0.365 | 0.392 | 0.366 | 0.394 | 0.390 | 0.404 | 0.395 | 0.407 | 0.426 | 0.420 | 0.424 | 0.415 | 0.399 | 0.410 | 0.410 | 0.411 |
| | 720 | 0.416 | 0.419 | 0.410 | 0.420 | 0.427 | 0.431 | 0.417 | 0.421 | 0.454 | 0.441 | 0.452 | 0.441 | 0.491 | 0.459 | 0.487 | 0.450 | 0.454 | 0.439 | 0.478 | 0.450 |
| | Avg | 0.344 | 0.380 | 0.346 | 0.382 | 0.354 | 0.384 | 0.352 | 0.383 | 0.381 | 0.395 | 0.382 | 0.397 | 0.407 | 0.410 | 0.414 | 0.407 | 0.387 | 0.400 | 0.406 | |
| ETTm2 | 96 | 0.169 | 0.259 | 0.167 | 0.257 | 0.175 | 0.265 | 0.173 | 0.262 | 0.175 | 0.258 | 0.171 | 0.256 | 0.180 | 0.264 | 0.182 | 0.265 | 0.175 | 0.259 | 0.187 | 0.267 |
| | 192 | 0.225 | 0.297 | 0.227 | 0.303 | 0.243 | 0.316 | 0.229 | 0.301 | 0.237 | 0.299 | 0.237 | 0.299 | 0.250 | 0.309 | 0.246 | 0.304 | 0.241 | 0.302 | 0.249 | 0.309 |
| | 336 | 0.277 | 0.331 | 0.285 | 0.346 | 0.294 | 0.343 | 0.286 | 0.341 | 0.298 | 0.340 | 0.296 | 0.338 | 0.311 | 0.348 | 0.307 | 0.342 | 0.305 | 0.343 | 0.321 | 0.351 |
| | 720 | 0.362 | 0.386 | 0.368 | 0.398 | 0.389 | 0.410 | 0.378 | 0.401 | 0.392 | 0.396 | 0.392 | 0.394 | 0.412 | 0.407 | 0.407 | 0.398 | 0.402 | 0.400 | 0.408 | 0.403 |
| | Avg | 0.258 | 0.318 | 0.262 | 0.326 | 0.275 | 0.334 | 0.266 | 0.326 | 0.275 | 0.323 | 0.274 | 0.322 | 0.288 | 0.332 | 0.286 | 0.327 | 0.281 | 0.326 | 0.291 | 0.333 |
| Average | | 0.302 | 0.329 | 0.311 | 0.337 | 0.318 | 0.339 | 0.317 | 0.337 | 0.339 | 0.342 | 0.334 | 0.340 | 0.342 | 0.347 | 0.377 | 0.362 | 0.354 | 0.354 | 0.376 | 0.362 |

# E Full Results

Here, we provide the full results of each experiment listed in the main text, including long-term/short-term forecasting, few-shot/zero-shot learning, time series classification, and ablation study.

## E.1 Long-term Forecasting

Table 10 presents the detailed long-term forecasting results across four prediction horizons $\{96, 192, 336, 720\}$ on seven widely used datasets. S$^2$TS-LLM consistently outperforms the high-performing baselines, achieving an average reduction of 2.9% in MSE and 2.5% in MAE compared to the second-best method, S$^2$IP-LLM [7]. These results validate the effectiveness of our representational and contextual alignment between time series and natural language.

## E.2 Short-term Forecasting

Table 11 illustrates the detailed short-term forecasting results on six M4 datasets. In this setting, prediction horizons range from 6 to 48, with input lengths set to twice the horizon. While our S$^2$TS-LLM achieves the best performance among LLM-based methods, including S$^2$IP-LLM [7], Time-LLM [9], and GPT4TS [22], it performs slightly below the lightweight model TimeMixer [41]. This may be due to the relatively short input sequences (ranging from 12 to 96), which limit the ability of LLMs to effectively model long-range dependencies.

## E.3 Few-shot Learning

Tables 12 and 13 provide the detailed few-shot forecasting results trained on the first 5% and 10% of the data, respectively, across various prediction lengths. The proposed S$^2$TS-LLM consistently

Table 11: Full short-term forecasting results. The prediction horizons are in $[6, 48]$, and the last three rows are weighted averaged from all datasets under different sampling intervals. A lower value indicates better performance. Red denotes the best performance, while Blue indicates the second-best.

| | Methods | S²TS-LLM | S²IP-LLM | Time-LLM | GPT4TS | TimeMixer | TimeXer | iTransformer | RLinear | PatchTST | TimesNet |
|---|---|---|---|---|---|---|---|---|---|---|---|
| Year. | SMAPE | 13.306 | 13.413 | 13.750 | 15.110 | 13.206 | 13.370 | 13.652 | 13.617 | 13.477 | 15.378 |
| | MASE | 3.007 | 3.024 | 3.055 | 3.565 | 2.916 | 3.002 | 3.095 | 2.991 | 3.019 | 3.554 |
| | OWA | 0.785 | 0.792 | 0.805 | 0.911 | 0.776 | 0.787 | 0.807 | 0.793 | 0.792 | 0.918 |
| Quart. | SMAPE | 10.242 | 10.352 | 10.671 | 10.597 | 9.996 | 10.323 | 10.353 | 10.752 | 10.380 | 10.465 |
| | MASE | 1.209 | 1.228 | 1.276 | 1.253 | 1.166 | 1.221 | 1.209 | 1.313 | 1.233 | 1.227 |
| | OWA | 0.906 | 0.922 | 0.950 | 0.938 | 0.825 | 0.914 | 0.911 | 0.967 | 0.921 | 0.923 |
| Month. | SMAPE | 12.967 | 12.995 | 13.416 | 13.258 | 12.605 | 13.076 | 13.079 | 13.338 | 12.959 | 13.513 |
| | MASE | 0.975 | 0.970 | 1.045 | 1.003 | 0.919 | 0.977 | 0.974 | 1.027 | 0.970 | 1.039 |
| | OWA | 0.908 | 0.910 | 0.957 | 0.931 | 0.869 | 0.912 | 0.911 | 0.945 | 0.905 | 0.957 |
| Others. | SMAPE | 4.837 | 4.805 | 4.973 | 6.124 | 4.564 | 5.054 | 4.780 | 4.979 | 4.952 | 6.913 |
| | MASE | 3.310 | 3.247 | 3.412 | 4.116 | 3.115 | 3.373 | 3.231 | 3.332 | 3.347 | 4.507 |
| | OWA | 1.031 | 1.017 | 1.053 | 1.259 | 0.982 | 1.064 | 1.012 | 1.049 | 1.049 | 1.438 |
| Avg. | SMAPE | 11.979 | 12.021 | 12.494 | 12.690 | 11.723 | 12.082 | 12.142 | 12.363 | 12.059 | 12.880 |
| | MASE | 1.610 | 1.612 | 1.731 | 1.808 | 1.559 | 1.621 | 1.631 | 1.662 | 1.623 | 1.836 |
| | OWA | 0.862 | 0.857 | 0.913 | 0.940 | 0.840 | 0.874 | 0.869 | 0.890 | 0.869 | 0.955 |

Table 12: Full few-shot learning results on 5% training data. We use the same protocol as in Table 10.

| Methods | | S²TS-LLM | | S²IP-LLM | | Time-LLM | | GPT4TS | | TimeMixer | | TimeXer | | iTransformer | | RLinear | | PatchTST | | TimesNet | |
|---|---|---|---|---|---|---|---|---|---|---|---|---|---|---|---|---|---|---|---|---|---|---|
| Metric | | MSE | MAE | MSE | MAE | MSE | MAE | MSE | MAE | MSE | MAE | MSE | MAE | MSE | MAE | MSE | MAE | MSE | MAE | MSE | MAE |
| ETTh1 | 96 | 0.482 | 0.447 | 0.500 | 0.493 | 0.518 | 0.498 | 0.543 | 0.506 | 0.558 | 0.502 | 0.496 | 0.469 | 0.808 | 0.610 | 0.501 | 0.474 | 0.557 | 0.519 | 0.892 | 0.625 |
| | 192 | 0.529 | 0.485 | 0.690 | 0.539 | 0.702 | 0.547 | 0.748 | 0.580 | 0.764 | 0.607 | 0.724 | 0.609 | 0.928 | 0.658 | 0.677 | 0.566 | 0.711 | 0.570 | 0.940 | 0.665 |
| | 336 | 0.730 | 0.577 | 0.761 | 0.620 | 0.725 | 0.603 | 0.754 | 0.595 | 0.855 | 0.644 | 0.934 | 0.662 | 1.475 | 0.861 | 0.876 | 0.638 | 0.816 | 0.619 | 0.945 | 0.653 |
| | 720 | - | - | - | - | - | - | - | - | - | - | - | - | - | - | - | - | - | - | - | - |
| | Avg | 0.580 | 0.503 | 0.650 | 0.550 | 0.648 | 0.549 | 0.681 | 0.560 | 0.726 | 0.584 | 0.718 | 0.580 | 1.070 | 0.710 | 0.685 | 0.559 | 0.695 | 0.569 | 0.925 | 0.647 |
| ETTh2 | 96 | 0.303 | 0.347 | 0.363 | 0.409 | 0.384 | 0.420 | 0.376 | 0.421 | 0.369 | 0.393 | 0.334 | 0.369 | 0.397 | 0.427 | 0.344 | 0.377 | 0.401 | 0.421 | 0.409 | 0.420 |
| | 192 | 0.360 | 0.395 | 0.375 | 0.411 | 0.394 | 0.424 | 0.418 | 0.441 | 0.451 | 0.439 | 0.461 | 0.444 | 0.438 | 0.445 | 0.505 | 0.459 | 0.452 | 0.455 | 0.483 | 0.464 |
| | 336 | 0.419 | 0.441 | 0.403 | 0.421 | 0.416 | 0.433 | 0.408 | 0.439 | 0.492 | 0.479 | 0.755 | 0.603 | 0.631 | 0.553 | 0.538 | 0.493 | 0.464 | 0.469 | 0.499 | 0.479 |
| | 720 | - | - | - | - | - | - | - | - | - | - | - | - | - | - | - | - | - | - | - | - |
| | Avg | 0.360 | 0.394 | 0.380 | 0.413 | 0.398 | 0.426 | 0.400 | 0.433 | 0.437 | 0.437 | 0.517 | 0.472 | 0.488 | 0.475 | 0.462 | 0.443 | 0.439 | 0.448 | 0.463 | 0.454 |
| ETTm1 | 96 | 0.440 | 0.423 | 0.357 | 0.390 | 0.422 | 0.424 | 0.386 | 0.405 | 0.563 | 0.484 | 0.509 | 0.462 | 0.589 | 0.510 | 0.426 | 0.425 | 0.399 | 0.414 | 0.606 | 0.518 |
| | 192 | 0.434 | 0.422 | 0.432 | 0.434 | 0.448 | 0.440 | 0.440 | 0.438 | 0.579 | 0.490 | 0.538 | 0.472 | 0.703 | 0.565 | 0.455 | 0.440 | 0.441 | 0.436 | 0.681 | 0.539 |
| | 336 | 0.455 | 0.440 | 0.440 | 0.442 | 0.452 | 0.447 | 0.485 | 0.459 | 0.599 | 0.510 | 0.597 | 0.512 | 0.898 | 0.641 | 0.514 | 0.479 | 0.499 | 0.467 | 0.786 | 0.597 |
| | 720 | 0.568 | 0.502 | 0.593 | 0.521 | 0.585 | 0.491 | 0.577 | 0.499 | 0.886 | 0.648 | 0.831 | 0.637 | 0.948 | 0.671 | 0.707 | 0.565 | 0.767 | 0.587 | 0.796 | 0.593 |
| | Avg | 0.474 | 0.447 | 0.455 | 0.446 | 0.477 | 0.451 | 0.472 | 0.450 | 0.657 | 0.533 | 0.619 | 0.521 | 0.784 | 0.596 | 0.526 | 0.477 | 0.526 | 0.476 | 0.717 | 0.561 |
| ETTm2 | 96 | 0.197 | 0.280 | 0.197 | 0.278 | 0.205 | 0.277 | 0.199 | 0.280 | 0.199 | 0.282 | 0.203 | 0.287 | 0.265 | 0.339 | 0.193 | 0.273 | 0.206 | 0.288 | 0.220 | 0.299 |
| | 192 | 0.257 | 0.319 | 0.254 | 0.322 | 0.267 | 0.336 | 0.256 | 0.316 | 0.262 | 0.320 | 0.265 | 0.325 | 0.310 | 0.362 | 0.265 | 0.319 | 0.264 | 0.324 | 0.311 | 0.361 |
| | 336 | 0.302 | 0.344 | 0.315 | 0.350 | 0.309 | 0.347 | 0.318 | 0.353 | 0.331 | 0.361 | 0.331 | 0.362 | 0.373 | 0.399 | 0.323 | 0.352 | 0.334 | 0.367 | 0.338 | 0.366 |
| | 720 | 0.414 | 0.410 | 0.421 | 0.421 | 0.448 | 0.432 | 0.460 | 0.436 | 0.502 | 0.463 | 0.480 | 0.445 | 0.478 | 0.454 | 0.461 | 0.435 | 0.454 | 0.432 | 0.509 | 0.465 |
| | Avg | 0.293 | 0.338 | 0.296 | 0.342 | 0.307 | 0.348 | 0.308 | 0.346 | 0.324 | 0.357 | 0.320 | 0.355 | 0.356 | 0.388 | 0.311 | 0.345 | 0.314 | 0.352 | 0.344 | 0.372 |
| Average | | 0.427 | 0.421 | 0.445 | 0.438 | 0.458 | 0.444 | 0.465 | 0.447 | 0.605 | 0.511 | 0.606 | 0.509 | 0.675 | 0.542 | 0.496 | 0.456 | 0.494 | 0.461 | 0.612 | 0.509 |

outperforms the baselines on most datasets, with the exception of ETTm1. Given the inherent sparsity and variability in few-shot learning scenarios, such performance differences on specific datasets are within a reasonable and acceptable range. Additionally, the results show that almost all LLM-based methods outperform conventional specialized models, highlighting the strong few-shot learning capabilities of large language models.

## E.4 Zero-shot Learning

Table 14 reports the detailed results of the zero-shot forecasting task across various prediction lengths. S²TS-LLM almost achieves the best performance across all settings, delivering a notable 8.3% improvement over the second-best method, RLinear. Interestingly, RLinear performs surprisingly well, possibly because its simple linear structure mitigates overfitting to the source dataset, enhancing generalization in the zero-shot setting. Different from RLinear, our S²TS-LLM effectively activates the internal knowledge of LLMs to facilitate cross-domain adaptation from source to target datasets.

## E.5 Time Series Classification

Table 15 presents the comparative results on 10 multivariate UEA classification datasets. For the time series classification task, we include a diverse set of baselines tailored for time series classification, including classical models (XGBoost [56], Rocket [55]), RNN-based models (LSSL [54], LSTNet [2]), Transformer-based models (Transformer [51], Informer [50], Non-stationary

Table 13: Full few-shot learning results on 10% training data. We use the same protocol as in Table 10.

| Methods | | S²TS-LLM | | S²IP-LLM | | Time-LLM | | GPT4TS | | TimeMixer | | TimeXer | | iTransformer | | RLinear | | PatchTST | | TimesNet | |
|---|---|---|---|---|---|---|---|---|---|---|---|---|---|---|---|---|---|---|---|---|---|
| Metric | | MSE | MAE | MSE | MAE | MSE | MAE | MSE | MAE | MSE | MAE | MSE | MAE | MSE | MAE | MSE | MAE | MSE | MAE | MSE | MAE |
| ETTh1 | 96 | 0.469 | 0.444 | 0.481 | 0.474 | 0.720 | 0.533 | 0.458 | 0.456 | 0.513 | 0.477 | 0.483 | 0.461 | 0.790 | 0.586 | 0.447 | 0.442 | 0.516 | 0.485 | 0.861 | 0.628 |
| | 192 | 0.493 | 0.472 | 0.518 | 0.491 | 0.747 | 0.545 | 0.570 | 0.516 | 0.571 | 0.507 | 0.549 | 0.481 | 0.837 | 0.609 | 0.544 | 0.492 | 0.598 | 0.524 | 0.797 | 0.593 |
| | 336 | 0.533 | 0.494 | 0.664 | 0.570 | 0.793 | 0.551 | 0.608 | 0.535 | 0.713 | 0.570 | 0.716 | 0.570 | 0.780 | 0.575 | 0.671 | 0.559 | 0.657 | 0.550 | 0.941 | 0.648 |
| | 720 | 0.678 | 0.560 | 0.711 | 0.584 | 0.880 | 0.584 | 0.725 | 0.591 | 1.004 | 0.717 | 0.991 | 0.697 | 1.234 | 0.811 | 0.839 | 0.636 | 0.762 | 0.610 | 0.877 | 0.641 |
| | Avg | 0.543 | 0.493 | 0.593 | 0.529 | 0.785 | 0.553 | 0.590 | 0.525 | 0.700 | 0.568 | 0.685 | 0.552 | 0.910 | 0.645 | 0.625 | 0.532 | 0.633 | 0.542 | 0.869 | 0.628 |
| ETTh2 | 96 | 0.305 | 0.349 | 0.354 | 0.400 | 0.334 | 0.381 | 0.331 | 0.374 | 0.338 | 0.365 | 0.330 | 0.365 | 0.404 | 0.435 | 0.328 | 0.362 | 0.353 | 0.389 | 0.378 | 0.409 |
| | 192 | 0.353 | 0.391 | 0.401 | 0.423 | 0.430 | 0.438 | 0.402 | 0.411 | 0.405 | 0.406 | 0.412 | 0.410 | 0.470 | 0.474 | 0.420 | 0.418 | 0.403 | 0.414 | 0.490 | 0.467 |
| | 336 | 0.392 | 0.420 | 0.442 | 0.450 | 0.449 | 0.458 | 0.406 | 0.433 | 0.475 | 0.463 | 0.498 | 0.467 | 0.489 | 0.485 | 0.516 | 0.476 | 0.426 | 0.441 | 0.537 | 0.494 |
| | 720 | 0.437 | 0.446 | 0.480 | 0.486 | 0.485 | 0.490 | 0.449 | 0.464 | 0.605 | 0.539 | 0.654 | 0.564 | 0.593 | 0.538 | 0.587 | 0.517 | 0.477 | 0.480 | 0.510 | 0.491 |
| | Avg | 0.372 | 0.402 | 0.419 | 0.439 | 0.424 | 0.441 | 0.397 | 0.421 | 0.456 | 0.446 | 0.474 | 0.451 | 0.489 | 0.483 | 0.463 | 0.443 | 0.415 | 0.431 | 0.479 | 0.465 |
| ETTm1 | 96 | 0.432 | 0.421 | 0.388 | 0.401 | 0.412 | 0.422 | 0.390 | 0.404 | 0.552 | 0.479 | 0.492 | 0.461 | 0.709 | 0.556 | 0.398 | 0.411 | 0.410 | 0.419 | 0.583 | 0.501 |
| | 192 | 0.436 | 0.421 | 0.422 | 0.421 | 0.447 | 0.438 | 0.429 | 0.423 | 0.571 | 0.492 | 0.525 | 0.465 | 0.717 | 0.548 | 0.430 | 0.426 | 0.437 | 0.434 | 0.630 | 0.528 |
| | 336 | 0.460 | 0.431 | 0.456 | 0.430 | 0.497 | 0.465 | 0.469 | 0.439 | 0.538 | 0.483 | 0.538 | 0.483 | 0.735 | 0.575 | 0.464 | 0.446 | 0.476 | 0.454 | 0.725 | 0.568 |
| | 720 | 0.530 | 0.472 | 0.554 | 0.490 | 0.594 | 0.521 | 0.569 | 0.498 | 0.687 | 0.534 | 0.671 | 0.515 | 0.752 | 0.584 | 0.528 | 0.472 | 0.681 | 0.556 | 0.769 | 0.549 |
| | Avg | 0.465 | 0.436 | 0.455 | 0.435 | 0.487 | 0.461 | 0.464 | 0.441 | 0.598 | 0.502 | 0.557 | 0.481 | 0.728 | 0.565 | 0.455 | 0.439 | 0.501 | 0.466 | 0.677 | 0.537 |
| ETTm2 | 96 | 0.188 | 0.272 | 0.192 | 0.274 | 0.224 | 0.296 | 0.188 | 0.269 | 0.189 | 0.269 | 0.196 | 0.279 | 0.245 | 0.322 | 0.183 | 0.260 | 0.191 | 0.274 | 0.212 | 0.285 |
| | 192 | 0.248 | 0.308 | 0.246 | 0.313 | 0.260 | 0.317 | 0.251 | 0.309 | 0.255 | 0.308 | 0.252 | 0.309 | 0.274 | 0.338 | 0.250 | 0.303 | 0.252 | 0.317 | 0.270 | 0.323 |
| | 336 | 0.295 | 0.338 | 0.301 | 0.340 | 0.312 | 0.349 | 0.307 | 0.346 | 0.310 | 0.345 | 0.318 | 0.351 | 0.361 | 0.394 | 0.313 | 0.343 | 0.306 | 0.353 | 0.323 | 0.353 |
| | 720 | 0.383 | 0.390 | 0.400 | 0.403 | 0.424 | 0.416 | 0.426 | 0.417 | 0.448 | 0.421 | 0.457 | 0.431 | 0.467 | 0.442 | 0.433 | 0.413 | 0.433 | 0.427 | 0.474 | 0.449 |
| | Avg | 0.279 | 0.327 | 0.284 | 0.332 | 0.305 | 0.344 | 0.293 | 0.335 | 0.301 | 0.336 | 0.306 | 0.343 | 0.336 | 0.373 | 0.295 | 0.330 | 0.296 | 0.343 | 0.320 | 0.353 |
| Average | | 0.415 | 0.415 | 0.438 | 0.434 | 0.500 | 0.450 | 0.436 | 0.431 | 0.562 | 0.478 | 0.513 | 0.460 | 0.616 | 0.570 | 0.460 | 0.436 | 0.461 | 0.446 | 0.586 | 0.496 |

Table 14: Full zero-shot learning results on ETT datasets. A lower value indicates better performance. Red denotes the best performance, while Blue indicates the second-best.

| Methods | | S²TS-LLM | | S²IP-LLM | | Time-LLM | | GPT4TS | | TimeMixer | | TimeXer | | iTransformer | | RLinear | | PatchTST | | TimesNet | |
|---|---|---|---|---|---|---|---|---|---|---|---|---|---|---|---|---|---|---|---|---|---|
| Metric | | MSE | MAE | MSE | MAE | MSE | MAE | MSE | MAE | MSE | MAE | MSE | MAE | MSE | MAE | MSE | MAE | MSE | MAE | MSE | MAE |
| ETTh1 ↓ ETTh2 | 96 | 0.279 | 0.339 | 0.315 | 0.377 | 0.324 | 0.368 | 0.335 | 0.374 | 0.297 | 0.346 | 0.308 | 0.351 | 0.353 | 0.394 | 0.296 | 0.342 | 0.304 | 0.350 | 0.358 | 0.387 |
| | 192 | 0.336 | 0.378 | 0.402 | 0.407 | 0.398 | 0.396 | 0.412 | 0.417 | 0.376 | 0.390 | 0.391 | 0.404 | 0.437 | 0.445 | 0.383 | 0.394 | 0.386 | 0.400 | 0.427 | 0.429 |
| | 336 | 0.362 | 0.402 | 0.453 | 0.432 | 0.410 | 0.423 | 0.441 | 0.420 | 0.411 | 0.420 | 0.439 | 0.436 | 0.482 | 0.476 | 0.420 | 0.428 | 0.414 | 0.428 | 0.449 | 0.451 |
| | 720 | 0.387 | 0.427 | 0.442 | 0.451 | 0.403 | 0.449 | 0.438 | 0.452 | 0.410 | 0.432 | 0.421 | 0.437 | 0.556 | 0.506 | 0.420 | 0.438 | 0.419 | 0.443 | 0.448 | 0.458 |
| | Avg | 0.341 | 0.387 | 0.403 | 0.417 | 0.384 | 0.409 | 0.406 | 0.422 | 0.374 | 0.397 | 0.390 | 0.407 | 0.457 | 0.455 | 0.380 | 0.401 | 0.380 | 0.405 | 0.421 | 0.431 |
| ETTh1 ↓ ETTm2 | 96 | 0.209 | 0.300 | 0.242 | 0.319 | 0.236 | 0.320 | 0.236 | 0.315 | 0.228 | 0.311 | 0.233 | 0.313 | 0.247 | 0.319 | 0.218 | 0.300 | 0.215 | 0.304 | 0.239 | 0.313 |
| | 192 | 0.260 | 0.330 | 0.286 | 0.337 | 0.265 | 0.353 | 0.287 | 0.342 | 0.278 | 0.337 | 0.286 | 0.343 | 0.293 | 0.350 | 0.277 | 0.335 | 0.275 | 0.339 | 0.291 | 0.342 |
| | 336 | 0.312 | 0.360 | 0.351 | 0.367 | 0.337 | 0.376 | 0.341 | 0.374 | 0.341 | 0.375 | 0.364 | 0.386 | 0.364 | 0.419 | 0.335 | 0.368 | 0.334 | 0.373 | 0.342 | 0.371 |
| | 720 | 0.394 | 0.409 | 0.422 | 0.416 | 0.429 | 0.430 | 0.435 | 0.422 | 0.436 | 0.424 | 0.437 | 0.423 | 0.534 | 0.470 | 0.432 | 0.420 | 0.431 | 0.424 | 0.434 | 0.419 |
| | Avg | 0.294 | 0.350 | 0.325 | 0.360 | 0.317 | 0.370 | 0.325 | 0.363 | 0.321 | 0.362 | 0.330 | 0.366 | 0.360 | 0.390 | 0.316 | 0.356 | 0.314 | 0.360 | 0.327 | 0.361 |
| ETTh2 ↓ ETTh1 | 96 | 0.509 | 0.478 | 0.668 | 0.567 | 0.618 | 0.515 | 0.732 | 0.577 | 0.476 | 0.459 | 0.507 | 0.477 | 0.854 | 0.606 | 0.455 | 0.444 | 0.485 | 0.465 | 0.848 | 0.601 |
| | 192 | 0.543 | 0.507 | 0.575 | 0.526 | 0.715 | 0.570 | 0.758 | 0.559 | 0.765 | 0.595 | 0.591 | 0.522 | 0.863 | 0.615 | 0.513 | 0.477 | 0.565 | 0.509 | 0.860 | 0.610 |
| | 336 | 0.558 | 0.517 | 0.655 | 0.577 | 0.636 | 0.523 | 0.759 | 0.578 | 0.678 | 0.558 | 0.705 | 0.579 | 0.867 | 0.626 | 0.606 | 0.524 | 0.581 | 0.515 | 0.867 | 0.626 |
| | 720 | 0.590 | 0.551 | 0.778 | 0.568 | 0.683 | 0.553 | 0.781 | 0.597 | 0.985 | 0.703 | 0.606 | 0.541 | 0.887 | 0.654 | 0.729 | 0.597 | 0.628 | 0.561 | 0.887 | 0.648 |
| | Avg | 0.550 | 0.513 | 0.669 | 0.560 | 0.663 | 0.540 | 0.757 | 0.578 | 0.726 | 0.579 | 0.602 | 0.530 | 0.868 | 0.625 | 0.576 | 0.511 | 0.565 | 0.513 | 0.865 | 0.621 |
| ETTh2 ↓ ETTm1 | 96 | 0.204 | 0.297 | 0.221 | 0.303 | 0.258 | 0.326 | 0.253 | 0.329 | 0.224 | 0.306 | 0.238 | 0.318 | 0.244 | 0.330 | 0.225 | 0.310 | 0.226 | 0.309 | 0.248 | 0.324 |
| | 192 | 0.262 | 0.332 | 0.295 | 0.344 | 0.303 | 0.342 | 0.293 | 0.346 | 0.312 | 0.368 | 0.293 | 0.351 | 0.291 | 0.356 | 0.288 | 0.347 | 0.289 | 0.345 | 0.296 | 0.352 |
| | 336 | 0.307 | 0.359 | 0.340 | 0.376 | 0.356 | 0.383 | 0.347 | 0.376 | 0.351 | 0.386 | 0.407 | 0.422 | 0.351 | 0.391 | 0.350 | 0.384 | 0.348 | 0.379 | 0.353 | 0.383 |
| | 720 | 0.394 | 0.419 | 0.453 | 0.428 | 0.440 | 0.434 | 0.446 | 0.429 | 0.517 | 0.490 | 0.462 | 0.447 | 0.452 | 0.451 | 0.452 | 0.441 | 0.439 | 0.427 | 0.471 | 0.446 |
| | Avg | 0.292 | 0.350 | 0.327 | 0.363 | 0.339 | 0.371 | 0.335 | 0.370 | 0.351 | 0.388 | 0.350 | 0.385 | 0.335 | 0.382 | 0.329 | 0.371 | 0.325 | 0.365 | 0.342 | 0.376 |
| ETTm1 ↓ ETTh2 | 96 | 0.327 | 0.375 | 0.358 | 0.382 | 0.355 | 0.403 | 0.353 | 0.392 | 0.382 | 0.406 | 0.385 | 0.408 | 0.371 | 0.407 | 0.350 | 0.383 | 0.354 | 0.385 | 0.377 | 0.407 |
| | 192 | 0.396 | 0.414 | 0.454 | 0.444 | 0.449 | 0.450 | 0.443 | 0.437 | 0.474 | 0.452 | 0.499 | 0.470 | 0.463 | 0.458 | 0.438 | 0.428 | 0.447 | 0.434 | 0.471 | 0.453 |
| | 336 | 0.408 | 0.431 | 0.488 | 0.452 | 0.479 | 0.467 | 0.469 | 0.461 | 0.585 | 0.523 | 0.609 | 0.529 | 0.481 | 0.485 | 0.463 | 0.463 | 0.481 | 0.463 | 0.472 | 0.484 |
| | 720 | 0.423 | 0.449 | 0.469 | 0.478 | 0.477 | 0.476 | 0.466 | 0.468 | 0.496 | 0.482 | 0.627 | 0.548 | 0.503 | 0.482 | 0.459 | 0.459 | 0.474 | 0.471 | 0.495 | 0.482 |
| | Avg | 0.389 | 0.417 | 0.442 | 0.439 | 0.440 | 0.449 | 0.433 | 0.439 | 0.484 | 0.466 | 0.530 | 0.489 | 0.455 | 0.458 | 0.428 | 0.431 | 0.439 | 0.438 | 0.457 | 0.454 |
| ETTm1 ↓ ETTm2 | 96 | 0.177 | 0.261 | 0.203 | 0.299 | 0.218 | 0.271 | 0.217 | 0.294 | 0.198 | 0.275 | 0.199 | 0.277 | 0.219 | 0.305 | 0.197 | 0.269 | 0.195 | 0.271 | 0.222 | 0.295 |
| | 192 | 0.237 | 0.303 | 0.272 | 0.325 | 0.288 | 0.335 | 0.277 | 0.327 | 0.263 | 0.316 | 0.269 | 0.324 | 0.277 | 0.347 | 0.258 | 0.307 | 0.258 | 0.311 | 0.288 | 0.337 |
| | 336 | 0.289 | 0.337 | 0.303 | 0.347 | 0.322 | 0.355 | 0.331 | 0.360 | 0.328 | 0.360 | 0.337 | 0.366 | 0.354 | 0.378 | 0.317 | 0.344 | 0.317 | 0.348 | 0.341 | 0.367 |
| | 720 | 0.368 | 0.386 | 0.436 | 0.418 | 0.414 | 0.409 | 0.429 | 0.413 | 0.423 | 0.409 | 0.436 | 0.419 | 0.426 | 0.420 | 0.415 | 0.399 | 0.416 | 0.404 | 0.436 | 0.418 |
| | Avg | 0.268 | 0.323 | 0.304 | 0.347 | 0.311 | 0.343 | 0.313 | 0.348 | 0.303 | 0.340 | 0.310 | 0.347 | 0.319 | 0.363 | 0.297 | 0.330 | 0.296 | 0.334 | 0.322 | 0.354 |
| ETTm2 ↓ ETTh2 | 96 | 0.298 | 0.357 | 0.324 | 0.383 | 0.334 | 0.416 | 0.360 | 0.401 | 0.325 | 0.370 | 0.337 | 0.378 | 0.347 | 0.401 | 0.334 | 0.376 | 0.327 | 0.367 | 0.360 | 0.401 |
| | 192 | 0.366 | 0.399 | 0.403 | 0.422 | 0.439 | 0.441 | 0.434 | 0.437 | 0.417 | 0.422 | 0.414 | 0.420 | 0.438 | 0.444 | 0.421 | 0.423 | 0.411 | 0.418 | 0.434 | 0.437 |
| | 336 | 0.403 | 0.426 | 0.434 | 0.442 | 0.455 | 0.457 | 0.460 | 0.459 | 0.437 | 0.444 | 0.444 | 0.448 | 0.459 | 0.464 | 0.459 | 0.454 | 0.439 | 0.447 | 0.460 | 0.459 |
| | 720 | 0.419 | 0.448 | 0.462 | 0.467 | 0.488 | 0.479 | 0.485 | 0.477 | 0.475 | 0.479 | 0.457 | 0.466 | 0.485 | 0.477 | 0.454 | 0.460 | 0.459 | 0.470 | 0.485 | 0.477 |
| | Avg | 0.372 | 0.408 | 0.406 | 0.429 | 0.429 | 0.448 | 0.435 | 0.443 | 0.414 | 0.429 | 0.413 | 0.428 | 0.432 | 0.447 | 0.409 | 0.425 | 0.409 | 0.425 | 0.435 | 0.443 |
| ETTm2 ↓ ETTm1 | 96 | 0.448 | 0.426 | 0.583 | 0.524 | 0.488 | 0.445 | 0.747 | 0.558 | 0.488 | 0.444 | 0.609 | 0.500 | 0.619 | 0.564 | 0.472 | 0.447 | 0.491 | 0.437 | 0.747 | 0.558 |
| | 192 | 0.471 | 0.446 | 0.609 | 0.501 | 0.555 | 0.464 | 0.781 | 0.560 | 0.527 | 0.465 | 0.677 | 0.536 | 0.685 | 0.565 | 0.519 | 0.467 | 0.530 | 0.470 | 0.781 | 0.560 |
| | 336 | 0.495 | 0.467 | 0.585 | 0.522 | 0.608 | 0.538 | 0.778 | 0.578 | 0.572 | 0.486 | 0.708 | 0.558 | 0.792 | 0.578 | 0.531 | 0.477 | 0.565 | 0.497 | 0.778 | 0.578 |
| | 720 | 0.568 | 0.508 | 0.712 | 0.579 | 0.699 | 0.566 | 0.769 | 0.573 | 0.635 | 0.528 | 0.846 | 0.620 | 0.727 | 0.579 | 0.585 | 0.506 | 0.686 | 0.565 | 0.769 | 0.573 |
| | Avg | 0.496 | 0.462 | 0.622 | 0.532 | 0.588 | 0.503 | 0.769 | 0.567 | 0.556 | 0.481 | 0.710 | 0.554 | 0.706 | 0.572 | 0.527 | 0.474 | 0.568 | 0.492 | 0.769 | 0.567 |
| Average | | 0.375 | 0.401 | 0.437 | 0.431 | 0.434 | 0.429 | 0.472 | 0.441 | 0.441 | 0.430 | 0.454 | 0.438 | 0.492 | 0.462 | 0.409 | 0.413 | 0.412 | 0.417 | 0.492 | 0.451 |

Transformer [48], FlowFormer [49]), an MLP-based model (LightTS [52]), and a CNN-based model (TCN [53]). Compared with these baselines across various architectures, our S²TS-LLM achieves superior performance, demonstrating its strong versatility and representational capacity.

Table 15: Full results for the classification task. '.' indicates the name of *former. A higher value indicates better performance. Red denotes the best performance, while Blue indicates the second-best.

| Methods | S²TS-LLM (Ours) | LLMs | | | CNNs | | MLPs | | | Transformers | | | | | | | RNNs | | Classical Methods | |
|---|---|---|---|---|---|---|---|---|---|---|---|---|---|---|---|---|---|---|---|---|
| | | S²IP-LLM | Time-LLM | GPT4TS | TimesNet | TCN | TimeMixer | Rlinear | LightTS | TimeXer | iTrans. | PatchTST | Flow. | Non-Station. | In. | Trans. | LSSL | LSTNet | Rocket | XGBoost |
| EthanolConcentration | 44.1 | 35.3 | 34.6 | 34.2 | 35.7 | 28.9 | 25.9 | 28.5 | 29.7 | 28.9 | 32.3 | 25.1 | 33.8 | 32.7 | 31.6 | 32.7 | 31.1 | 39.9 | 45.2 | 43.7 |
| FaceDetection | 69.7 | 68.5 | 67.9 | 69.2 | 68.6 | 52.8 | 67.3 | 63.4 | 67.5 | 69.4 | 68.5 | 66.2 | 67.6 | 68.0 | 67.0 | 67.3 | 66.7 | 65.7 | 64.7 | 63.3 |
| Handwriting | 34.2 | 33.1 | 32.0 | 32.7 | 32.1 | 53.3 | 25.3 | 22.6 | 26.1 | 24.6 | 31.7 | 26.8 | 33.8 | 31.6 | 32.8 | 32.0 | 24.6 | 25.8 | 58.8 | 15.8 |
| Heartbeat | 79.5 | 77.5 | 78.0 | 77.2 | 78.0 | 75.6 | 74.6 | 64.4 | 75.1 | 74.6 | 75.6 | 72.2 | 77.6 | 73.7 | 80.5 | 76.1 | 72.7 | 77.1 | 75.6 | 73.2 |
| JapaneseVowels | 98.9 | 98.6 | 98.1 | 98.6 | 98.4 | 98.9 | 95.9 | 96.5 | 96.2 | 97.8 | 98.3 | 95.7 | 98.9 | 99.2 | 98.9 | 98.7 | 98.4 | 98.1 | 96.2 | 86.5 |
| PEMS-SF | 90.7 | 88.4 | 87.2 | 87.9 | 89.6 | 68.8 | 87.9 | 86.1 | 88.4 | 79.8 | 88.4 | 74.6 | 83.8 | 87.3 | 81.5 | 82.1 | 86.1 | 86.7 | 75.1 | 98.3 |
| SelfRegulationSCP1 | 93.5 | 91.4 | 92.8 | 93.2 | 91.8 | 84.6 | 92.5 | 86.7 | 89.8 | 87.0 | 90.7 | 81.9 | 92.5 | 89.4 | 90.1 | 92.2 | 90.8 | 84.0 | 90.8 | 84.6 |
| SelfRegulationSCP2 | 62.2 | 58.3 | 57.2 | 59.4 | 57.2 | 55.6 | 56.7 | 52.8 | 51.1 | 53.3 | 56.6 | 53.3 | 56.1 | 57.2 | 53.3 | 53.9 | 52.2 | 52.8 | 53.3 | 48.9 |
| SpokenArabicDigits | 99.8 | 99.9 | 99.5 | 99.2 | 99.2 | 95.6 | 99.1 | 95.9 | 100.0 | 98.5 | 100.0 | 97 | 98.8 | 100.0 | 100.0 | 98.4 | 100.0 | 100.0 | 71.2 | 69.6 |
| UWaveGestureLibrary | 91.9 | 88.7 | 89.3 | 88.1 | 85.3 | 88.4 | 83.4 | 75.6 | 80.3 | 84.1 | 82.5 | 81.6 | 86.6 | 87.5 | 85.6 | 85.6 | 85.9 | 87.8 | 94.4 | 75.9 |
| Average | 76.4 | 73.9 | 73.7 | 74.0 | 73.6 | 70.3 | 70.9 | 67.2 | 70.4 | 69.8 | 72.5 | 67.4 | 73.0 | 72.7 | 72.1 | 71.9 | 70.9 | 71.8 | 72.5 | 66.0 |

Table 16: Full MSE results of ablation Study on the ETTh1 and ETTh2 datasets. A lower value indicates better performance. Red denotes the best performance, while Blue indicates the second-best.

| Dataset | | S²TS-LLM (Ours) | (Q1) Component Contribution | | | (Q2) Scalability | | (Q3) LLM Effectiveness | | |
|---|---|---|---|---|---|---|---|---|---|---|
| | | | w/o Symbol | w/o Segment | w/o Fusion | w Vocab | w BERT | w/o Pretrain | w Att | w Trans |
| ETTh1 | 96 | 0.355±0.009 | 0.361±0.001 | 0.361±0.001 | 0.356±0.003 | 0.351±0.003 | 0.360±0.001 | 0.368±0.002 | 0.388±0.004 | 0.384±0.003 |
| | 192 | 0.399±0.001 | 0.403±0.001 | 0.403±0.001 | 0.401±0.005 | 0.398±0.003 | 0.398±0.003 | 0.429±0.001 | 0.420±0.007 | 0.411±0.006 |
| | 336 | 0.402±0.004 | 0.432±0.015 | 0.431±0.005 | 0.415±0.003 | 0.401±0.005 | 0.404±0.002 | 0.452±0.005 | 0.437±0.012 | 0.433±0.004 |
| | 720 | 0.463±0.001 | 0.477±0.026 | 0.468±0.003 | 0.479±0.015 | 0.464±0.001 | 0.467±0.006 | 0.488±0.007 | 0.489±0.030 | 0.467±0.020 |
| | Avg | 0.404±0.003 | 0.418±0.010 | 0.416±0.002 | 0.413±0.002 | 0.403±0.002 | 0.407±0.002 | 0.434±0.002 | 0.432±0.009 | 0.424±0.005 |
| ETTh2 | 96 | 0.260±0.003 | 0.274±0.004 | 0.275±0.003 | 0.269±0.005 | 0.261±0.002 | 0.257±0.001 | 0.275±0.002 | 0.298±0.002 | 0.291±0.003 |
| | 192 | 0.312±0.004 | 0.335±0.005 | 0.338±0.004 | 0.319±0.004 | 0.311±0.002 | 0.313±0.001 | 0.347±0.004 | 0.362±0.006 | 0.364±0.001 |
| | 336 | 0.322±0.001 | 0.336±0.001 | 0.339±0.001 | 0.321±0.003 | 0.321±0.004 | 0.319±0.001 | 0.360±0.001 | 0.366±0.004 | 0.359±0.003 |
| | 720 | 0.392±0.002 | 0.406±0.005 | 0.408±0.006 | 0.408±0.005 | 0.394±0.004 | 0.394±0.004 | 0.403±0.004 | 0.426±0.016 | 0.419±0.008 |
| | Avg | 0.321±0.002 | 0.345±0.003 | 0.347±0.003 | 0.329±0.003 | 0.321±0.003 | 0.321±0.001 | 0.346±0.003 | 0.362±0.005 | 0.356±0.001 |

## E.6 Ablation Study

We conduct an ablation study with eight model variants on the ETTh1 and ETTh2 datasets to address the following questions: **(Q1)** *Do individual components contribute meaningfully to the framework?* **(Q2)** *How scalable is the framework?* and **(Q3)** *Are LLMs effective for time series tasks?* Table 16 reports the detailed ablation results using the MSE metric, averaged over three runs.

To answer Q1, we design the following three variants of S²TS-LLM, including "w/o Symbol", "w/o Segment", and "w/o Fusion":

- **"w/o Symbol":** Removes the spectral symbolization component (Section 3.2), meaning the model does not learn textual abstractions to align time series with natural language in the representation space.

- **"w/o Segment":** Removes the contextual segmentation component (Section 3.3), meaning the model does not learn segment-level positions for aligning temporal structures with natural language contexts.

- **"w/o Fusion":** Replaces our time-linguistic fusion strategy (Section 3.4) with a simple summation of the temporal and textual LLM outputs.

As shown in Table 16, removing either the spectral symbolization or contextual segmentation leads to performance drops, as they disrupt the knowledge transferring from natural language to time series. This highlights the necessity of both components in enabling LLMs to comprehend temporal semantics and structural relationships. Additionally, the inferior performance of "w/o Fusion" variant indicates the effectiveness of our fusion strategy in bridging the representational gap between time series and natural language.

Regarding Q2, we design two variants of S²TS-LLM, including "w Vocab" and "w BERT", to evaluate the scalability of the symbolic representation and the adaptability of different LLM architectures:

- **"w Vocab":** Replaces ASCII symbols with vocabulary for textual abstracting. Specifically, we query LLMs to select 24 vocabulary terms that are well-suited for describing frequency characteristics, including trend, daily, harmonic, peak, rhythm, cycle, noise, disturbance, pattern, pulse, beat, fluctuation, seasonality, vibration, waveform, instability, perturbation, anomaly, impulse, oscillation, turbulence, interference, variation, and echo.

- **"w BERT":** Substitutes GPT-2 with BERT [47] as the backbone LLM. Both GPT-2 and BERT have comparable sizes (12 Transformer layers, 124M vs. 110M parameters). Following our GPT-2 setup, we utilize the first 4 layers of BERT for time series analysis.

As shown in Table 16, the **"w Vocab"** setting can yield accurate and stable predictions, because vocabulary-based representations describe frequency patterns more precisely than ASCII symbols. While the "w Vocab" variant uses general vocabulary terms for frequency descriptions, we believe that customizing the word set for specific datasets could further enhance performance. However, this approach requires prior knowledge and could introduce substantial preprocessing overhead as the vocabulary size grows. Additionally, the **"w BERT"** variant achieves competitive results relative to the default GPT-2 used in $S^2$TS-LLM, demonstrating the scalability and flexibility of our framework.

Finally, to address Q3, we design three variants by removing the LLM from our framework:

- **"w/o Pretrain":** Initializes the LLM parameters randomly, without any pre-training.
- **"w Att":** Replaces the GPT-2 backbone with a single Attention layer.
- **"w Trans":** Substitutes the GPT-2 backbone with a single Transformer layer.

All three variants discard the prior knowledge of pre-trained LLMs, resulting in the poorest performance among all settings. These results highlight the importance of leveraging LLMs for time series analysis, as their ability to model long-range dependencies and their rich internal knowledge facilitate the effective extraction of periodic patterns and comprehension of multimodal data [20].

