# OpenReview forum: "Bridging Time and Linguistics: LLMs as Time Series Analyzer through Symbolization and Segmentation"
_NeurIPS.cc/2025/Conference — NeurIPS 2025 poster_

### Official Review · Reviewer_Ttyn · 2025-06-22

**Clarity:** 2
**Significance:** 2
**Originality:** 2
**Rating:** 3
**Confidence:** 5

**Summary:**

The paper proposes a framework to repurpose large language models for time series analysis by focusing on the context gaps between natural language and time series data at the representation level. It introduces two core components, including a spectral symbolization that converts time series into sparse frequency-domain representations and then map them to ASCII symbols, and contextual segmentation that reassigns positional indices using GRU to capture temporal continuity of patched inputs. Experiments across multiple tasks demonstrate the effectiveness of the proposed method, including short-term and long-term forecasting, few-shot/zero-shot learning, and classification.

**Questions:**

1) Could the authors provide the evaluation of related LLM-based time series forecasting methods (mentioned above), to validate the claims in terms of outperforming state-of-the-art baselines?

2) Could the author justify the motivation of using ASCII symbols for frequency component tokenization?

3) In the methodology design, why the frequency-domain output in equation 8 only serves a gating function instead of an actual input to be fused?

**Ethical Concerns:**

["NO or VERY MINOR ethics concerns only"]

**Final Justification:**

My concerns persist regarding the fact that, after extensive modeling stages for the gated residual component, it is further controlled by a cosine-similarity score, which blurs the clarity of its individual contribution and motivation. The additional results (including the comparison with state-of-the-art method) has reduced some of my initial concerns, and I am less negative toward this paper.

**Limitations:**

Yes

**Quality:**

3

**Strengths And Weaknesses:**

Strengths:

1. The high-level idea of using Fourier analysis in time series tokenization is somewhat  interesting. The idea of using an auto-regressive model to capture patched input for positional encoding is also well-motivated and interesting.

2. The proposed method is evaluated across a wide range of tasks, demonstrating certain advantages over the provided baselines.


Weaknesses:

1. This paper still lacks comparisons and discussions with several recent and relevant baselines in LLM-based time series analysis, particularly those focused on context modeling, such as TimeCMA [1] and FSCA [2].

2. The ablation study only focuses on long-term time series forecasting, which seems insufficient to demonstrate the effectiveness of the provided component, especially for Q1.  Results of short-term and classification tasks would be needed for a more comprehensive assessment.

3. The authors stated the disadvantages of time-domain tokenization with an example in lines 162-164, which seems not quite clear and precise. It would be better if a frequency-domain example is provided. Moreover, in line 182, the motivation of using ASCII symbols for frequency component tokenization is not clear, more insights will be helpful.

4. In equation 8, it seems that the frequency-domain output *$O^{n}_{T}$* that passes through a sigmoid function only serves as a gating function, in addition to a cosine similarity for weighted fusion. This design is a bit confusing as *$O^{n}_{T}$* is never used as an actual input for fusion.

[1] TimeCMA: Towards LLM-Empowered Multivariate Time Series Forecasting via Cross-Modality Alignment

[2] Context-Alignment: Activating and Enhancing LLMs Capabilities in Time Series

---

> ### Author Rebuttal · Authors · 2025-07-31
>
> ## W1, Q1: More Discussion and Comparison of Related LLM-based Methods
>
> 1. **Discussion**
>
>     A: We will include more discussions on LLM-based related works in the revised version. Specifically, TimeCMA aims to enhance forecasting performance by leveraging LLM-generated text as an augmentation to time series data. To this end, it retrieves and selects textual information highly relevant to the time series, constructing disentangled and robust cross-modal embeddings for prediction. Unlike TimeCMA, which uses LLMs as external modules, our S$^2$TS-LLM tokenizes time series into textual descriptions, providing prompts to directly help LLMs better capture temporal dynamics during forecasting. FSCA addresses the cross-modal alignment problem in LLMs by representing time series and textual prompts as hierarchical graph nodes. It constructs a directed graph to characterize structural relationships between time series and prompts, and then uses graph convolution to fuse their representations for forecasting. In contrast, our S$^2$TS-LLM does not model the relationship between time series and textual prompts. Instead, we captures internal temporal context by reassigning the position indices of time-series patches, making the time-series representation more akin to linguistic context and thus easier for LLMs to interpret.
>
> 2. **Comparison**
>
>     A: In light of your suggestions, we add TimeCMA for further long-term forecasting comparison, as listed below. Although both our method and TimeCMA achieve cross-modal alignment, they differ in direction: TimeCMA adopts an LLMs-for-Time-Series strategy, using LLM-generated text to augment time-series features without adapting LLMs to temporal understanding. In contrast, we adopt a Time-Series-for-LLMs strategy that transforms time series into  language-like representations by tokenizing frequency patterns and learning temporal context. This bridges the gaps between LLMs and time series, leading to improved performance of our method.
>
>     |Method|Metric|Weather|ECL|ETTh1|ETTh2|ETTm1|ETTm2|
>     |:-:|:-:|:-:|:-:|:-:|:-:|:-:|:-:|
>     |TimeCMA|MSE↓|0.250|0.174|0.423|0.372|0.380|0.275|
>     ||MAE↓|0.276|0.269|0.431|0.397|0.392|0.323|
>     |S$^2$TS-LLM(Ours)|MSE↓|0.225|0.164|0.404|0.321|0.344|0.258|
>     ||MAE↓|0.265|0.256|0.429|0.376|0.380|0.318|
>
> ## W2: More Ablation Study of Short-term Forecasting and Classification Tasks
>
> A: To further evaluate the capability of designed components, we conduct additional ablation study on short-term forecasting and classification tasks as follows. The results consistently show that each module contributes positively across different tasks. Notably, although reduced input length in short-term forecasting limits available frequency information, our method still effectively distinguishes temporal patterns by dynamically selecting top-$k$ components, providing valuable prompts to guide LLMs in forecasting. For classification, contextual segmentation proves most effective, likely because it transforms raw signals into discrete semantic units by reassigning index, making LLMs  easier to capture local patterns and improving performance.
>
> |Short-term Forecasting|Metric|S$^2$TS-LLM(Ours)|w/o Symbolization|w/o Segmentation|w/o Fusion|
> |:-:|:-:|:-:|:-:|:-:|:-:|
> |M4-Year.|SMAPE↓|13.306|13.718|13.680|13.537|
> ||MASE↓|3.007|3.092|3.109|3.049|
> ||OWA↓|0.785|0.809|0.810|0.789|
> |M4-Quart.| SMAPE↓|10.242|10.303|10.435|10.529|
> ||MASE↓|1.209|1.219|1.239|1.254|
> ||OWA↓|0.906|0.912|0.926|0.935|
>
> |Classification|Metric|S$^2$TS-LLM (Ours)|w/o Symbolization|w/o Segmentation|w/o Fusion|
> |:-:|:-:|:-:|:-:|:-:|:-:|
> |Heartbeat|Acc↑|79.5|77.6|77.1|78.1|
> |FaceDetection|Acc↑|70.3|68.3|67.9|68.6|
>
> ## W3, Q2: Clarify Tokenization Example and Motivation of Using ASCII Symbol
>
> 1. **Clarify Tokenization Example**
>
>     A: The example of time-domain tokenization in lines 162-164 aims to convey a message similar to the motivation illustrated in Fig. 1(a), but its simplified expression may lead to ambiguity. To clarify, consider the sequence $[66, 45, 24, 32, 27, 33, 16, 8]$. A time-domain tokenization that captures only the global trend might represent this sequence as "decreasing from 66 to 8," thereby omitting intermediate fluctuations such as 24→32→27→33. To provide a clearer description, more words are required. For example, GPT-3.5 generates: "The series starts at 66 and drops quickly to 24 over the first three points. After reaching 24, it briefly rises to 32 and 33, then drops again to 16 and finally 8." However, such a verbose textual description significantly exceeds the length of the original sequence. Using such long-sequence prompts to train LLMs for time series forecasting would result in significant computational overhead, making it impractical. Therefore, tokenizing a time series for prompting LLMs requires striking a balance between accuracy and efficiency, which is highly challenging.
>
> 2. **More Explanation of Frequency-domain Tokenization**
>
>     A: Compared to time-domain tokenization, representing compact frequency-domain information in textual form is more tractable. For the same sequence $[66, 45, 24, 32, 27, 33, 16, 8]$, applying a real-valued FFT yields the frequency-domain components $[251+0j, 30.5-33.4j, 53.0-38.0j, 47,5-17.5j, 15+0j]$, which respectively represents the DC component and the 1/8 to 4/8 frequency bands. By selecting the top 2 components with the largest magnitudes for reconstruction, we obtain the sequence $[56.5, 35.6, 13.8, 33.4, 32.8, 46.2, 22.5, 10.4]$. Although the reconstructed values are not numerically accurate, the sequence successfully preserves the three key trends of the original: an initial decline, a fluctuation, and a subsequent drop. This demonstrates that the temporal pattern of a sequence can be effectively captured by tokenizing only a small number of dominant frequency components. In doing so, frequency-domain tokenization mitigates the limitations of time-domain approaches, which may introduce noise when overly concise or lead to high computational costs when overly verbose.
>
> 3. **The Motivation of Using ASCII Symbols**
>
>     A: The motivation for using ASCII symbols to tokenize frequency components lies in the observation that different temporal patterns can be effectively characterized by combinations of salient frequency components. By assigning each frequency component a unique ASCII symbol, we can represent the top-k components of a sequence as a compact symbolic string. For example, combinations dominated by low-frequency symbols typically indicate smooth and stable trends of time series, while those containing more high-frequency symbols suggest volatile or rapidly changing patterns of time series.
>
>     Although ASCII symbols do not carry semantic meaning in natural language, LLMs can still learn their distributional statistics after fine-tuning[1]. In this way, ASCII tokens function similarly to foreign-language or domain-specific tokens, allowing the LLMs to associate certain patterns of symbols with frequency characteristics. As shown in our ablation study (Section 4.6), using ASCII symbols to represent frequency components achieves comparable forecasting performance to using natural-language words, demonstrating their effectiveness as interpretable and learnable tokens for LLMs.
>
> > [1] W. J. Su, et al. Do Large Language Models (Really) Need Statistical Foundations? arXiv, 2025
>
> ## W4, Q3: Confusing Design of Time-Linguistic Fusion
>
> A: To clarify, our time-linguistic fusion (TLF) design is inspired by the Gated TCN module in GraphWavelet[2], which uses a gating mechanism of the form:
> $h = g(\Theta \star \mathcal{X} + b) \odot \sigma (\Theta \star \mathcal{X} + c)$. The sigmoid function $\sigma(\cdot)$ controls the proportion of information from $\mathcal{X}$ that is propagated to the next layer.
>
> In a similar spirit, we apply a sigmoid transformation to the frequency-domain output, which acts as a gating signal. This gate determines which parts of the time-series information should be preserved or suppressed during the fusion process. Following the gating, we incorporate cosine similarity to further modulate the fusion weights, emphasizing components of the time-domain signal that are more semantically aligned with the frequency-domain patterns.
>
> We intentionally do not fuse the raw frequency-domain output directly with the time-domain representation, as the prediction task is inherently defined in the time domain. The frequency-domain output is instead treated as a complementary signal that guides the fusion process without shifting the prediction space. To evaluate the effectiveness of our fusion mechanism, we further conduct comparisons with several alternative fusion strategies, including bilinear fusion ($z = x^T W y$), gated fusion ($z = \alpha x + (1 - \alpha)y$), and attention-based fusion. The following experimental results show that our proposed TLF fusion achieves the best performance, consistently yielding lower prediction errors than the alternatives.
>
> |||TLF(ours)|Bilinear|Gate|Att|
> |:-:|:-:|:-:|:-:|:-:|:-:|
> ||Length|MSE↓, MAE↓|MSE↓, MAE↓|MSE↓, MAE↓|MSE↓ , MAE↓|
> |ETTh1|96|0.355, 0.387|0.359, 0.389|0.367, 0.396|0.369, 0.397|
> ||192|0.399, 0.419|0.412, 0.423|0.402, 0.422|0.401, 0.421|
> ||336|0.402, 0.432|0.439, 0.441|0.423, 0.443|0.446, 0.463|
> ||720|0.463, 0.478|0.462, 0.473|0.482, 0.485|0.502, 0.497|
> ||AVG|0.405, 0.429|0.418, 0.431|0.418, 0.436|0.430, 0.444|
> |ETTh2|96|0.260, 0.329|0.276, 0.335|0.273, 0.334|0.291, 0.345|
> ||192|0.312, 0.363|0.338, 0.375|0.335, 0.377|0.340, 0.379|
> ||336|0.322, 0.380|0.336, 0.385|0.333, 0.386|0.348, 0.398|
> ||720|0.392, 0.430|0.393, 0.428|0.400, 0.432|0.406, 0.439|
> ||AVG|0.322, 0.376|0.336, 0.381|0.335, 0.382|0.346, 0.390|
>
> > [2] Z. H. Wu, et al. Graph WaveNet for Deep Spatial-Temporal Graph Modeling. IJCAI, 2019

---

> > ### Comment · Reviewer_Ttyn · 2025-08-04
> >
> > Thanks for the response. The presented results in W2, L1 resolve my concerns in ablation for other claimed task settings. However, I still have remaining concerns and questions regarding the results and justifications.
> >
> > Baselines. FSCA is an important and generally relevant LLM-based baseline with state-of-the-art performance, in the scope of this paper, regarding context alignment illustrated in Figure 2 of the author’s manuscript. The comparison would still be needed to validate the author’s claim of outperforming state-of-the-arts.
> >
> > The motivation of ASCII symbols. The use of ASCII symbols to encode frequency components results in a discrete pattern space, which appears used purely as pattern markers in the embedding space. LLMs with a large semantic space are not inherently necessary for modeling discrete frequency patterns unless there is meaningful symbolic structure or linguistic context. Based on Figure 6 and appendix, the variant with vocabulary also outperforms the ASCII one.
> >
> > Fusion. Graph Wavenet performs self-gating for prediction, the strategy (self-gated by time-domain outputs) could be compared as well. The attention-based mechanism needs more details (e.g., self-attention or cross-attention). The general concern here is that, after extensive modeling stages detailed in Sections 3.2 and 3.4, it only serves as a gated residual connection that is additionally controlled by a cosine-similarity score, which blurs the actual contribution of the ASCII-encoded frequency-domain component.

---

> > > ### Author Response · Authors · 2025-08-06
> > >
> > > ## Q1 (Baselines): More Comparison with FSCA
> > >
> > > A: Per to your suggestion, we reproduce the FSCA for comparison. We notice that FSCA did not switch the LLM backbone to evaluation mode during validation, potentially introducing randomness (e.g., from dropout[1]) that could bias checkpoint selection and test results. To ensure fairness, we follow standard practice by enabling evaluation mode during validation and strictly adhering to FSCA's reported hyperparameters for reproduction.
> > >
> > > The results are listed below. While our S$^2$TS-LLM and FSCA respectively adopt different forms of context alignment (intra-modal context vs. inter-modal context), both show the value of contextual cues in LLM-based time-series modeling. However, unlike FSCA, which alters the LLMs' architecture by adding a GNN layer for alignment, our method retains the original LLM structure by externally reassigning positional indices. This offers a more flexible solution for cross-modal adaptation.
> > >
> > > |Method|Metric|Weather|Traffic|ETTh1|ETTh2|ETTm1|ETTm2|
> > > |:-:|:-:|:-:|:-:|:-:|:-:|:-:|:-:|
> > > |FSCA|MSE↓|0.230|0.397|0.414|0.325|0.348|0.261|
> > > ||MAE↓|0.265|0.274|0.437|0.379|0.384|0.323|
> > > |S$^2$TS-LLM(Ours)|MSE↓|0.225|0.396|0.404|0.321|0.344|0.258|
> > > ||MAE↓|0.265|0.278|0.429|0.376|0.380|0.318|
> > >
> > > > [1] Srivastava. Dropout: A Simple Way to Prevent Neural Networks from Overfitting. JMLR, 2014
> > >
> > > ## Q2 (The motivation of ASCII symbols): More Justification for Using ASCII Symbols for Tokenization
> > >
> > > A: Thank you for the insightful comment. Since LLMs are pre-trained not only on natural language but also on symbolic forms, ASCII symbols can provide semantic and even structural cues to LLMs. For example, SepLLM[2] shows that certain ASCII symbols, such as separators, often receive higher attention scores than content words in LLMs, indicating that such symbols can represent inter-sentence semantics. ArtPrompt[3] also shows that ASCII-based drawings (e.g., using "*" to mimic banned word "bomb") can bypass LLMs’ content filters, eliciting restricted responses (e.g., answer how to make a bomb). These findings suggest that ASCII symbols, despite lacking explicit semantics, carry representational weight within LLMs.
> > >
> > > We emphasize that ASCII tokenization is not central to our method as it is not necessarily required. As shown in our ablation, lexical tokenization also works well and even improves robustness. ASCII is presented as a flexible alternative for general-purpose tokenizing, which avoids selecting task- or data-specific vocabulary.
> > >
> > > > [2] Chen. SepLLM:Accelerate Large Language Models by Compressing One Segment into One Separator. ICML, 2025
> > > [3] Jiang. ArtPrompt: ASCII Art-based Jailbreak Attacks against Aligned LLMs. ACL, 2024

---

> > > ### Author Response · Authors · 2025-08-06
> > >
> > > ## Q3 (Fusion): More Justification about Fusion's Effect on Frequency-domain Tokenization
> > >
> > > 1. More Comparison with Self-gating Mechanism
> > >
> > >     A: Per your suggestion, we add a comparison with a self-gating variant that fuses only time-domain outputs, i.e., $O_X^{(n)} \odot \sigma \left ( O_X^{(n)} \right )$. While self-gating performs single-view fusion within the time domain, our Time-Linguistics Fusion (TLF) integrates frequency and time-domain features from a multi-view perspective. The superior results of TLF over self-gating highlights the benefit of multi-view fusion, which aligns with findings in prior representation learning research [4].
> > >
> > >     |||ETTh1|||||ETTh2|||||
> > >     |:-:|:-:|:-:|:-:|:-:|:-:|:-:|:-:|:-:|:-:|:-:|:-:|
> > >     ||Length|96|192|336|720|AVG|96|192|336|720|AVG|
> > >     |Self-Gating|MSE↓|0.368|0.403|0.445|0.490|0.427|0.271|0.337|0.341|0.406|0.339|
> > >     ||MAE↓|0.400|0.426|0.462|0.496|0.446|0.335|0.379|0.395|0.444|0.388|
> > >     |TLF(Ours)|MSE↓|0.355|0.399|0.402|0.463|0.405|0.260|0.312|0.322|0.392|0.322|
> > >     ||MAE↓|0.387|0.419|0.432|0.478|0.429|0.329|0.363|0.380|0.430|0.376|
> > >
> > > 2. Will Fusion Affect the Contribution of Frequency-domain Tokenization?
> > >
> > >     A: Thanks for your constructive comment. We respectfully clarify that our Time-Linguistics Fusion (TLF) module does not diminish the contribution of the frequency-domain component for two main reasons:
> > >
> > >     - **The fusion mechanism preserves frequency information.**
> > >         Based on the definition of sigmoid function ($\frac{1}{1 + e^{-x}}$), our gated fusion design softly modulates frequency information by smoothly mapping features into a new embedding space within the range (0, 1), rather than discarding it. This aligns with the findings in gated architectures like LSTM and GRU: the LSTM[5] update rule, $h_t = \text{Sigmoid}(W h_{t-1} + b) \odot \text{tanh}(C_t)$, illustrates that sigmoid gating continuously propagates information from the previous state $h_{t-1}$ to the next state $h_t$.
> > >
> > >     - **Residual strategy is more effective for fusing frequency cues.**
> > >         While residual connections may reduce the relative importance of frequency features, they are well-suited for integrating auxiliary frequency cues with raw numerical features. This is because frequency features guide LLMs semantically rather then numerically. Thus, residual pathways offers smoother integration without disrupting the original signal structure. As shown in our ablations, it consistently outperforms direct connections such as bilinear, gating, and attention-based fusion, confirming its effectiveness.
> > >
> > >     To further validate our approach, we evaluate its magnitude sensitivity and trend preservation from a frequency-domain perspective. We compare it against direct connection variants and representative frequency-based models (FEDformer[6] and FiLM[7]). For magnitude sensitivity, we compute the L2 distance between the frequency components of predictions and ground truth. For trend preservation, we measure the dominant frequency overlap to assess how well each model captures underlying top-k frequency trends. Results show that our method outperforms both the direct connection variants and the frequency-based baselines, indicating the effectiveness of our method in preserving frequency-domain contributions for time-series modeling.
> > >
> > >     ||||TLF|Bilinear|Gate|Att|FEDformer|FiLM|
> > >     |:-:|:-:|:-:|:-:|:-:|:-:|:-:|:-:|:-:|
> > >     |ETTh1|Magtitude Test|L2 Distance↓|22.44|23.89|23.58|23.68|23.01|23.33|
> > >     ||Trend Test|Dominant Frequency Overlap↑|0.523|0.518|0.509|0.506|0.503|0.507|
> > >     |ETTh2|Magtitude Test|L2 Distance↓|23.28|24.50|24.16|24.73|31.08|37.39|
> > >     ||Trend Test|Dominant Frequency Overlap↑|0.851|0.848|0.848|0.842|0.844|0.846
> > >
> > > 3. More Details of Attention-based Fusion
> > >
> > >     A: We implement Attention-based Fusion using a standard cross-attention mechanism, where frequency-domain features serve as the query and time-domain features as the key and value: $\text{Softmax}\left ( \frac{(O_T^{(n)} W_Q) (O_X^{(n)} W_K)^T}{\sqrt d} \right ) \left (O_X^{(n)} W_V \right )$.
> > >
> > > > [4] Li. A Survey of Multi-view Representation Learning. TKDE, 2018
> > > [5] Hochreiter. Long Short-term Memory. NC, 1997
> > > [6] Zhou. Fedformer: Frequency Enhanced Decomposed Transformer for Long-term Series Forecasting. ICML, 2022
> > > [7] Zhou. FiLM: Frequency Improved Legendre Memory Model for Long-term Time Series Forecasting. NIPS, 2022

---

> > > > ### Comment · Reviewer_Ttyn · 2025-08-08
> > > >
> > > > Thank you for your time and detailed rebuttal. I have no further questions for my assessment and will take these points into account in my final evaluation.

---

> > > > > ### Author Response · Authors · 2025-08-09
> > > > >
> > > > > We are greatly encouraged by your willingness to take these points into consideration in your final evaluation. Thank you once again for your time and thoughtful feedback.

---

### Official Review · Reviewer_HLD5 · 2025-06-30

**Clarity:** 2
**Significance:** 2
**Originality:** 2
**Rating:** 4
**Confidence:** 4

**Summary:**

This paper aims to enhance the LLM’s perception of structures and patterns of time series data by bridging the modality gap between time series and linguistics. It proposes S2TS-LLM, a framework to repurpose LLM for time series analysis by (1) a spectral symbolization to endow time series with frequency-based textual abstractions (2) a contextual segmentation paradigm to mitigate the structural mismatch between time series and natural language. The proposed method is evaluated on five mainstream analysis tasks following the setting in GPT4TS.

**Questions:**

1. Both previous time-domain descriptions and Spectral Symbolization can provide information about time series patterns from different perspectives. Is Spectral Symbolization superior and can provide more information about time series variations than time-domain descriptions?
2. Why can ASCII symbols enhance the comprehension of spectral patterns by LLMs? Is there any evidence or literature to support this assertion? (**Lines 181-183**)
3. Contextual Segmentation groups the patches into blocks to assign grouped indices, which might destroy the temporal order of patches in positional encoding, thus hinder the LLM’s comprehension of structural continuity embedded in the temporal orders, for example, an upward trend patch and a subsequent downward patch construct a peak shape in time series. How to solve this new challenge brought by the proposed Contextual Segmentation? (**Lines 184-200**)
4. What is the relation between B segments and Q patches? (**Line 193**)
5. Since this work focuses on LLM for time series analysis, there have currently been many publications using LLM for time series. More experiments about LLM-based methods should be added to explain why the non-LLM baselines should be reduced. The relevant LLM baselines include TEST, CALF, UniTime, TEMPO, and so on.
6. Why choose GPT-2 as a backbone? For time-LLM, which uses LLaMA as a backbone, the comparison is not fair. Is there any backbone ablation study?

**Ethical Concerns:**

["NO or VERY MINOR ethics concerns only"]

**Final Justification:**

The authors’ thorough rebuttal has addressed most of my earlier concerns. While I still find the novelty to be somewhat limited, the methodology is sound and the empirical evaluation is solid. I've adjusted my score to 4.

**Limitations:**

yes.

**Paper Formatting Concerns:**

NA.

**Quality:**

3

**Strengths And Weaknesses:**

**Strengths**

1. The paper is well structured and the figures are rich and clear to convey related ideas.
2. The code is provided, and the experiments are extensive, including multiple task,s and the analysis of the core mechanisms of the model is sufficient.
3. The paper proposes to construct the textual description of each time series sample in the frequency domain, which is a novel perspective compared with previous works, like Time-LLM.

**Weaknesses**

1. The novelty of the paper is limited, as there have been several works focusing on bridging the modality gap between time series and language by aligning textual and temporal semantics, like Time-LLM, TEST, CALF [1], and S2 IP-LLM. Moreover, the author has not addressed the shortcomings of previous work, and the proposed innovations are not compelling.

2. The motivation and effectiveness of the proposed Spectral Symbolization are not convincingly established. (**Lines 43-61**)
    * The authors claim that existing time-domain descriptions fail to capture complex patterns in time series data and advocate for frequency-domain representations. While frequency-domain analysis can effectively capture periodic and steady-state features, it falls short in addressing "temporal localization" and "non-stationary changes," which are crucial for identifying time series patterns. Consequently, while frequency-domain descriptions may serve as valuable supplements, they are not inherently superior to time-domain analyses.
    * I am skeptical about the approach's ability to "losslessly capture the infinite variability of the time domain" and to "describe intricate variations with concise language." These objectives seem to be the core challenges that the paper aims to tackle, yet the proposed method does not convincingly demonstrate its effectiveness in achieving them.

3. The observation of assigning sequential indices to individual time points lacks literature support and seems to be imaginary. (**Lines 71-84**) The baselines, including Time-LLM, GPT4TS, and S2 IPLLM, all utilize a patching strategy, with positional encoding implemented at the patch level rather than the time point level. Patches can preserve local temporal context and model continuity in temporal trends within the patch. The author should compare their proposed segment-level positional encoding with the previous patch-level positional encoding, instead of time point-based token-level positional encoding.

[1] Peiyuan, Liu, et al., "CALF: Aligning LLMs for Time Series Forecasting via Cross-modal Fine-Tuning," AAAI 2025.

---

> ### Author Rebuttal · Authors · 2025-07-31
>
> ## W1:Limited Novelty
>
> A: We respectfully clarify that the novelty of our S$^2$TS-LLM lies in two aspects:
>
> - **Frequency-domain Tokenization**: We adapt LLMs to time-series tasks from a frequency-domain perspective, setting it apart from prior time-domain methods like Time-LLM, TEST, CALF, and S$^2$IP-LLM. Based on the insight that top-k frequency components can capture key temporal trends, we tokenize them as textual prompts to help LLMs in distinguish different time-series patterns. This dominant-frequency tokenization is conceptually akin to keypoint detection in CV, where a few salient features (e.g., tires) can represent the whole object (e.g., car). This makes our method be more flexible and efficient for cross-modal alignment than time-domain methods that rely on comparing token discrepancies between text and time series one by one.
>
> - **New Context Alignment**: We go beyond prior work by introducing a contextual alignment, which endows temporal segments with language-like contextual structures. This alignment is done externally by reassigning positional indices of time series, without altering LLM's internal architecture, making our design plug-and-play and more friendly for LLMs' cross-modal adaptation.
>
> ## W2,W1,Q1:Motivation, Challenge and Effectiveness of Spectral Symbolization (SS)
>
> 1. **Motivation of SS**
>
>     A: Our motivation is to leverage the sparsity of frequency signals to improve the clarity of textual prompts for LLMs. Prior studies [1][2] show that prompts are most effective when they are clear and precise, not necessarily long. Time-domain prompts, while rich in information, often require verbose language to detailedly express trends, local variations, etc., which may overwhelm or confuse LLMs. In contrast, frequency signals are often sparse and quantifiable, with key temporal features concentrated in a few dominant components (as shown in Fig. 1). This sparsity enables the generation of compact and unambiguous prompts that help LLMs distinguish global temporal patterns such as periodicity, volatility, or smoothness.
>
> 2. **Not Address the Core Challenge**
>
>     A: We respectfully clarify that our goal is not to "losslessly capture the infinite variability of the time domain," but to appropriately preserve essential patterns that support LLMs' cross-modal adaptation. The core challenge lies in the trade-off between accuracy and efficiency when tokenizing time series as prompts for LLMs. We address this by shifting tokenization from the time domain to the frequency domain.
>
>     To illustrate, consider tokenizing a sequence $[66,45,24,32,27,33,16,8]$:
>
>     - Time-domain Tokenization: When aiming for efficiency, it tends to oversimplify temporal dynamics (e.g., describing a sequence as "decreasing from 66 to 8"), which ignores local variations and thus lacks accuracy. When seeking for accuracy, it needs to capture fine-grained changes, which often requires lengthy descriptions and hurts efficiency. For example, a precise description-"The series starts at 66 and drops to 24 over the first three points. After reaching 24, it briefly rises to 32 and 33, then drops again to 16 and finally 8"-far exceeds the length of the original sequence.
>
>     - Frequency-domain Tokenization: Through FFT, we reduce the sequence length by half, improving the efficiency due to less information needed to be tokenized. Meanwhile, since using top-2 frequency component can reconstruct a sequence $[57,36,14,33,32,46,23,10]$ with original "drop→flucuation→drop" trends, our frequency-domain tokenization can preserve global patterns to ensure accuracy.
>
>     Based on these observation, our method can leverage the sparsity of frequency features to generate compact yet informative prompts, balancing accuracy and efficiency more effectively than time-domain tokenization.
>
> 3. **Effectivenss of SS**
>
>     A: While the frequency domain contains less information than the time domain, prior work [3] have shown that frequency features are effective for modeling time-series. These successes support the potential of frequency learning to address LLMs' cross-modal adaptation. Importantly, our method does not rely solely on frequency features. We combine frequency-domain prompts with raw time-domain series, allowing LLMs to benefit from complementary information: the frequency prompts helps LLMs perceive global trends, while raw series allows attention mechanisms to capture local variations. As shown in Section 4 and Appendix Table 7, our method outperforms time-domain methods, like Time-LLM and S$^2$IP-LLM, in both prediction accuracy and computational efficiency, highlighting the benefits of our method.
>
> > [1]Chen.Unleashing the potential of prompt engineering for large language models.arXiv
> [2]Kusano.Are Longer Prompts Always Better? Prompt Selection in Large Language Models for Recommendation Systems.arXiv
> [3]Zhou.Fedformer:Frequency enhanced decomposed transformer for long-term series forecasting.ICML
>
> ## W3,Q4:Point-level Mismatches Patch-level in Segmentation
>
> A: We believe that the term 'time points' may have unintentionally caused some misunderstanding. We appreciate the reviewer's observation and will revise ambiguous terms like "time points" to improve clarity. As stated in lines 147-150 and 188-191, our method, like Time-LLM, first applies a patching strategy to preprocess time series and then performs contextual segmentation at the patch level, not at individual time points. The detailed process is listed below:
>
> - A univariate time series $X \in \mathbb{R}^{L}$ is divided into $Q$ non-overlapping patches, resulting in $X \in \mathbb{R}^{Q \times M}$.
>
> - Contextual segmentation is performed over these $Q$ patches. It groups adjacent and semantically similar patches into the same segment for assigning a shared position index, with each segment composed of one or more consecutive patches. For example, GPT-2 assigns sequential indices $[0,1,2,3,4]$ to 5 patches, while our segmentation may group patches 0, 1, 2 into one segment, assigning indices $[0,0,0,1,2]$ to reflect contextual continiuty.
>
> - Segment-level indices are fed into the the LLM's positional encoding, allowing patches within the same segment to share positional embeddings. This helps the attention mechanism fuse related patches, better capturing local trend continuity.
>
> ## Q2:Why ASCII Symbol Works?
>
> A: Thank you for the insightful question. ASCII symbols help LLMs understand spectral patterns for two main reasons: 1) they offer a compact and discriminative encoding of frequency components, and 2) LLMs, as statistical models, can learn meaningful patterns from symbolic inputs even without explicit semantics.
>
> - Discriminative representation: By mapping each component to a unique ASCII symbol, dynamically selecting top-k components can form a specific symbolic sequences that differentiate between spectral patterns. For example, sequences dominated by low-frequency symbols reflect smooth trends, while high-frequency symbols indicate volatility.
>
> - Statistical nature of LLMs: LLMs rely on statistical patterns rather than semantic meaning[4]; for example, [5] shows ASCII separators act as structural cues in LLMs. Thus, LLMs can interpret ASCII symbols as frequency-related tokens (like foreign language) via fine-tuning.
>
> Our ablation (Section 4.6) further confirms that ASCII tokens perform comparably to natural language words, validating their utility in frequency-domain prompting.
>
> > [4]Su.Do Large Language Models (Really) Need Statistical Foundations?arXiv
> [5]Chen.SepLLM:Accelerate Large Language Models by Compressing One Segment into One Separator.ICML
>
> ## Q3:Contextual Segmentation may Destroy Temporal Order
>
> A: We respectfully clarify that our segmentation preserves the temporal order of patches. Only adjacent and semantically similar patches are grouped into the same segment, and later patches never get lower position indices than earlier ones, making the sequence in order. This ensures the LLMs perceive the correct sequence structures; for example, a peak (upward followed by downward), won't be mistaken for a valley.
>
> ## Q5:More LLM-based Comparison
>
> A: Per your suggestions, we add TEST and CALF as baselines for comparison. Both of them perform cross-modal alignment from the time domain. In contrast, our method achieves better results by using frequency-domain alignment, which guides LLMs through simpler and clearer prompts. Additionally, our contextual segmentation endows time series with language-like structure. Assume that time series can be treated as character-like sequence, $[x_0,x_1,x_2,x_3,x_4,x_5] =$ ["c","a","t","d","o","g"]. Our method assigns grouped indices [0,0,0,1,1,1], forming semantic units like "cat" and "dog." Compared to LLMs' default sequential indexing [0,1,2,3,4,5], this grouping helps preserve trend continuity, enabling LLMs better modeling of time-series patterns.
>
> |||Weather|ECL|Traffic|ETTh1|ETTh2|ETTm1|
> |:-:|:-:|:-:|:-:|:-:|:-:|:-:|:-:|
> |CALF|MSE↓|0.250|0.175|0.439|0.432|0.349|0.395|
> ||MAE↓|0.274|0.265|0.281|0.428|0.382|0.390|
> |TEST|MSE↓|0.229|0.162|0.430|0.414|0.331|0.353|
> ||MAE↓|0.271|0.253|0.295|0.431|0.380|0.382|
> |S$^2$TS-LLM(Ours)|MSE↓|0.225|0.164|0.396|0.404|0.321|0.344|
> ||MAE↓|0.265|0.256|0.278|0.429|0.376|0.380|
>
> ## Q6:Backbone Fairness
>
> A: We follow GPT4TS and S$^2$IP-LLM in using GPT-2 as the backbone. As noted in Appendix (lines 41-42), the Time-LLM results reported are also based on GPT-2, ensuring fair comparison. To verify the robustness of our method across backbones, our ablation (Fig. 6) has included a BERT-based variant. Here, we further compare with Time-LLM using LLaMA, and results show that our method achieves comparable performance with fewer layers (4 vs. 8), verifying its architectural robustness.
>
> |||ETTh1||ETTh2||
> |:-:|:-:|:-:|:-:|:-:|:-:|
> ||Length|96|192|96|192|
> |Time-LLM(8)|MSE↓|0.39|0.41|0.30|0.33|
> ||MAE↓|0.42|0.43|0.37|0.38|
> |S$^2$TS-LLM(4)|MSE↓|0.39|0.42|0.30|0.35|
> ||MAE↓|0.41|0.43|0.36|0.38|

---

> > ### Comment · Reviewer_HLD5 · 2025-08-05
> >
> > Thank you to the authors for your thorough response. Most of my concerns have been addressed. While I still find the novelty to be somewhat limited, I acknowledge the soundness of the methodology and the empirical efforts. I am willing to adjust my score accordingly.

---

> > > ### Author Response · Authors · 2025-08-05
> > >
> > > We are sincerely grateful for your valuable and constructive reviews, which have helped us further improve the quality of our paper. We are glad that our responses have addressed most of your concerns. Thank you again for your time and thoughtful feedback.

---

### Official Review · Reviewer_mbFi · 2025-07-02

**Clarity:** 3
**Significance:** 4
**Originality:** 3
**Rating:** 5
**Confidence:** 4

**Summary:**

This work proposes a model called S2TS-LLM, which repurposes LLMs for time series analysis. The method consists of two major components: (1) a spectral symbolization module that converts time series into the frequency-domain for textual abstraction; (2) a contextual segmentation module that segments time series and groups positional encodings based on the temporal patterns. The representations are later fed into a pre-trained LLM for downstream applications. Authors performed comprehensive experimental evaluation of the proposed method, showing that S2TS-LLM outperforms state-of-the-art methods across multiple standard time series tasks.

**Questions:**

Can the authors design synthetic experiments or design case studies to understand when will the proposed method have the most significant gains, and when will it fail?

**Ethical Concerns:**

["NO or VERY MINOR ethics concerns only"]

**Limitations:**

yes

**Quality:**

4

**Strengths And Weaknesses:**

Strength

1. Overall, this paper presents a creative methodology and a valid bunch of innovative contributions
- The spectral symbolization module emphasizes the frequency domain representation, which aligns with previous works [1, 2]
- The contextual segmentation module makes valid contribution in terms of emphasizing time series shape structures.
2. Comprehensive ablation experiments, demonstrating the effectiveness of the proposed components
- Table 6 is very informative.
3. Decent amount of experimental evaluation and informative interpretation of the results
4. Informative and insightful visualizations and analysis.

[1] Liu, Ran, Ellen L. Zippi, Hadi Pouransari, Chris Sandino, Jingping Nie, Hanlin Goh, Erdrin Azemi, and Ali Moin. "Frequency-aware masked autoencoders for multimodal pretraining on biosignals." arXiv preprint arXiv:2309.05927 (2023).

[2] Zhang, Xiang, Ziyuan Zhao, Theodoros Tsiligkaridis, and Marinka Zitnik. "Self-supervised contrastive pre-training for time series via time-frequency consistency." Advances in neural information processing systems 35 (2022): 3988-4003.

Weakness

1. The work could use more comprehensive experimental evaluation
- Can the proposed method be used on anomaly detection experiments, and how it would perform when comparing to other baselines?
- Why only performing classification experiments on 10 UEA datasets? Can the method perform well on all UEA and UCR datasets?
- When applying the method on different tasks and datasets, did the authors investigate at which condition, or on which dataset, the proposed method can perform significantly better than other methods? In other words, when would the method provide the biggest performance margin?
2. The work can include more comprehensive literature review

---

> ### Author Rebuttal · Authors · 2025-07-31
>
> ## W1, Q1: More Experimental evaluation
>
> 1. **Anomaly Detection**
>
>     A: We clarify that our S$^2$TS-LLM can be applied on anomaly detection tasks, and the results provided below showing its reasonable performance. However, S$^2$TS-LLM currently falls short of state-of-the-art models such as the LLM-based GPT4TS. One possible reason is that our top-k frequency selection mechanism is designed to capture dominant and generalizable patterns in the time series, rather than rare or subtle deviations. While this focus aligns well with forecasting or classification tasks, it may overlook the sparse and abnormal signals that are essential for anomaly detection. In future work, we plan to explore alternative selection strategies and anomaly-sensitive frequency encodings to better adapt our method to this task.
>
>     |F1-Score ↑ |S$^2$TS-LLM (Ours)|GPT4TS|PatchTST|ETS  |DLinear|
>     |:-:|:-:|:-:|:-:|:-:|:-:|
>     |SMD        |83.93             |86.89 |84.62   |83.13|77.10  |
>     |MSL        |82.16             |82.45 |78.70   |85.03|84.88  |
>     |SWaT       |92.65             |94.23 |85.72   |84.91|87.52  |
>     |PSM        |96.71             |97.13 |96.08   |91.76|93.55  |
>     |Average    |88.61             |90.18 |86.28   |86.21|85.76  |
>
> 2. **Experiments on all UEA Datasets**
>
>     A: Following prior work [1][2], we select 10 representative datasets from the UEA time-series classification archive to strike a balance between experimental diversity and computational feasibility, as LLM-based methods are relatively resource-intensive. These 10 datasets span a wide range of application domains, including biomedical signals (e.g., Heartbeat, SelfRegulationSCP1, SelfRegulationSCP2), chemical signals (e.g., EthanolConcentration), human activity (e.g., UWaveGestureLibrary, PEMS-SF), speech and audio (e.g., JapaneseVowels, SpokenArabicDigits), and visual tasks (e.g., FaceDetection, Handwriting). We believe this selection supports both comparability and meaningful evaluation.
>
> 3. **More Evaluation on UCR Dataset**
>
>     A: We believe that the results on the UEA datasets provide meaningful insights into the method's potential generalizability to the UCR archive. According to [3], the UEA archive is developed as an extension of the UCR archive, offering more challenging and diverse benchmarks with multivariate and longer time series, in contrast to the mostly univariate and short-sequence nature of UCR datasets.
>
> 4. **Study of Model's Performance Gain**
>
>     A: To investigate when our S$^2$STS-LLM works well and when it fails, we follow prior work [4][5] to evaluate model performance under different lookback lengths $\{512, 336, 192, 96, 48\}$. The detailed results are as follows.
>
>     We observe that S$^2$STS-LLM outperforms the other two LLM-based methods, S2IP-LLM and GPT4TS, when the lookback length is between 512 and 96. However, as the lookback length decreases to 48, S$^2$STS-LLM experiences a significant performance drop and underperforms compared to the S2IP-LLM and GPT4TS. This is consistent with our paper's observation that S$^2$STS-LLM achieves suboptimal results on short-term forecasting tasks. The main reason is that S$^2$STS-LLM uses FFT to extract dominant frequency components, which leads to further information compression when the lookback window is short. As a result, the generated compact textual descriptions may not provide LLMs with sufficient prompts to bridge the semantic gap between linguistics and time series.
>
>     Therefore, our method is better suited for long-sequence scenarios. For example, our model shows significant gains in both long-term forecasting and zero-shot learning tasks, where it can capture enough dominant frequency components to distinguish different temporal patterns.
>
>     |     |Lookback Length   | 512          | 336          | 192          | 96           | 48           |
>     |:-:|:-:|:-:|:-:|:-:|:-:|:-:|
>     |     |     Metric       | MSE↓ , MAE↓  | MSE↓ , MAE↓  | MSE↓ , MAE↓  | MSE↓ , MAE↓  | MSE↓ , MAE↓  |
>     |ETTh1|S$^2$TS-LLM (Ours)| 0.355, 0.387 | 0.365, 0.389 | 0.372, 0.391 | 0.378, 0.391 | 0.398, 0.411 |
>     |     |    S2IP-LLM      | 0.368, 0.403 | 0.367, 0.398 | 0.368, 0.393 | 0.379, 0.394 | 0.390, 0.401 |
>     |     |     GPT4TS       | 0.376, 0.397 | 0.378, 0.399 | 0.380, 0.401 | 0.383, 0.401 | 0.385, 0.403 |
>     |ETTh2|S$^2$TS-LLM (Ours)| 0.260, 0.329 | 0.281, 0.335 | 0.288, 0.338 | 0.291, 0.336 | 0.310, 0.350 |
>     |     |    S2IP-LLM      | 0.284, 0.345 | 0.282, 0.343 | 0.288, 0.343 | 0.293, 0.343 | 0.301, 0.344 |
>     |     |     GPT4TS       | 0.285, 0.342 | 0.286, 0.345 | 0.294, 0.349 | 0.295, 0.349 | 0.299, 0.350 |
>
> > [1] T. Zhou, et al. One fits all: Power general time series analysis by pretrained lm. NIPS, 2023
> [2] H. X. Wu, et al. Long. Timesnet: Temporal 2d-variation modeling for general time series analysis. ICLR, 2023
> [3] A. J. Bagnall, et al. The uea multivariate time series classification archive. arXiv, 2018
> [4] M. Jin, et al. Time-llm: Time series forecasting by reprogramming large language models. ICLR, 2024
> [5] W. Z. Yue, et al. FreEformer: Frequency Enhanced Transformer for Multivariate Time Series Forecasting. arXiv, 2025
>
> ## W2: More Literature Review
>
> A: Thanks for the comments. We will expand the discussion of related work by incorporating [6][7], to provide useful preliminaries on frequency-domain modeling and further demonstrate the potential of frequency-based representations in time-series applications with LLMs.
>
> > [6] R. Liu, et al. Frequency-aware masked autoencoders for multimodal pretraining on biosignals. arXiv 2023.
> [7] X. Zhang, et al. Self-supervised contrastive pre-training for time series via time-frequency consistency. NIPS, 2022

---

> > ### Comment · Reviewer_mbFi · 2025-08-05
> >
> > Thank you - I believe the new experiments made the work more complete. I'd remain my score unchanged.

---

> > > ### Author Response · Authors · 2025-08-06
> > >
> > > Thank you sincerely for your thoughtful feedback and for recognizing our work. Best wishes!

---

### Official Review · Reviewer_UTxa · 2025-07-03

**Clarity:** 3
**Significance:** 3
**Originality:** 3
**Rating:** 4
**Confidence:** 3

**Summary:**

The authors adapts GPT-2 to time-series tasks through two add-on modules, Spectral Symbolization and Contextual Segmentation.
SS turn patch into short string of ASCII symbols that stand for the *k* biggest FFT peaks.
CS use GRU patch embeddings, clustered them and assign unique positional ID.
They claim forecasting and classification show improved results compared to prior LLM for TS baselines.

**Questions:**

1. Does contextual segmentation really beat other long-context tricks (e.g., hierarchical attention, relative positions)? I only see the ablation without it.
2. Is the simple cosine-similarity gate the best way to fuse the raw-value and spectral streams?
3. After all, why should casting time-series into a handful of FFT-based ASCII symbols help a frozen GPT-2 transfer its linguistic priors? I don't see even a guess.

**Ethical Concerns:**

["NO or VERY MINOR ethics concerns only"]

**Limitations:**

I feel there are a handful of situations where SS will cause information loss, though they could perform well on some dataset.
1. Feed it x^2 or log⁡ x; the DC term plus a handful of low harmonics can’t recover the curvature, which can be easily described by text.
2. “Noisy but meaningful” beats. Heart-rate variability: the clinical signal sits in tiny shifts around 0.1 Hz, but the big energy spike is the main beat at ≈1 Hz.
3. Highly tortuous trajectories**.** A pedestrian GPS trace is full of zig-zags; FFT shoves all that into a smear of high-freq energy that says nothing about *where* the person is going.
SS is very much limited to wave-like time series, but in many real world applications time series.
M4 dataset is a good reference but it contains none of the above situations. The variability is very limited.

**Quality:**

3

**Strengths And Weaknesses:**

*Strength*

- The amount of empirical validation seems solid with 23 datasets and decent ablations
- Contextual segmentation is novel and well motivated. Shared positions solve long flat segment problem.
- Training budget is friendly. Backbone is frozen, only a few norms move.

*Weakness*

- Spectral symbolisation isn’t new. FreqTST (https://openreview.net/pdf?id=N1cjy5iznY) already tokenised FFT peaks. FreEformer (https://arxiv.org/pdf/2501.13989) works fully in frequency space. Though it's understandable that neither is cited because they are not peer reviewed.
- Evidence for “spectral has *better* information” is one ETTh1 ablation and a pretty heat-map; no magnitude-sensitive or trend-heavy test at all. I will explain why I suspect this idea in limitation.
- GPT-2 gets per-set LR sweeps while classic models use canned configs.

---

> ### Author Rebuttal · Authors · 2025-07-31
>
> ## W1:Lack Citations of Similar Idea of FreqTST and FreEformer
>
> A: We appreciate the reviewer's suggestion and will add a more detailed discussion of related frequency-domain works in the revised manuscript. While FreqTST and FreEformer all adopts frequency learning, our approach is conceptually and technically distinct from both:
>
> - FreEformer is a non-LLM-based method that focuses on improving the rank of attention matrices to better handle the sparsity in frequency data for forecasting. In contrast, our S$^2$TS-LLM, a LLM-based framework, aims to bridging the representational gap between time series and language for LLMs. Thus, our method transforms spectral signals to generate textual descriptions, prompting LLMs to capture time-series patterns for downstream tasks.
>
> - FreqTST aims to construct a new token dictionary that can generalizably represent frequency features across datasets to enhance Transformer-based forecasting. To achieve this, it designs a pretraining strategy to vectorize frequency amplitudes into token embeddings. In contrast, our method avoids creating a new dictionary, as LLMs struggle to understand newly introduced token vocabularies. Instead, we leverage the existing token vocabulary of LLMs to represent time series, enabling seamless and efficient adaptation to time-series tasks without additional pretraining. To this end, we propose a frequency-domain tokenization that maps each frequency component to a unique character and dynamically selects dominant frequency components to form distinctive textual descriptions for different time series. Our approach is conceptually akin to keypoint detection in CV, where identifying a few salient features (e.g., car tires) is sufficient to recognize the entire object (e.g., car).
>
> ## W2,L1:Potential Information Loss Problem of Spectral Symbolization Module
>
> A: We thank the reviewer for pointing out potential information loss problem of FFT in certain cases, such as curved trajectories. Although our Spectral Symbolization (SS) module is built upon FFT, its design helps mitigate this limitation. First, SS primarily uses FFT to generate textual descriptions of frequency-domain features, which serve as supplementary inputs alongside the original time-domain data for LLM training. Second, the SS module dynamically selects top-k frequency components for encoding, enabling it to capture discriminative patterns through varying combinations of dominant frequencies, even when the signal energy is concentrated in narrow frequency bands.
>
> Below, we provide an ablation study of short-term forecasting on the M4 datasets to support the effectiveness of our SS module. Although FFT intensifies the information compression problem in short-term forecasting, which may limit the relative contribution of the SS module compared to the other two modules, it still offers valuable frequency patterns that effectively help LLMs understand temporal dynamics.
>
> |||S$^2$TS-LLM(Ours)|w/o Spectral Symbolization|w/o Contextual Segmentation|w/o Fusion|
> |:-:|:-:|:-:|:-:|:-:|:-:|
> |M4-Year.|SMAPE↓|13.306|13.718|13.680|13.537|
> || MASE↓|3.007|3.092|3.109|3.049|
> ||OWA↓|0.785|0.809|0.810|0.789|
> |M4-Quart.|SMAPE↓|10.242|10.303|10.435|10.529|
> ||MASE↓|1.209|1.219|1.239|1.254|
> || OWA↓|0.906|0.912|0.926|0.935|
>
> Additionally, we further evaluate the magnitude sensitivity and trend preservation capabilities of our method, in comparison with two representative frequency-based models (FEDformer[1] and FiLM[2]). Regarding magnitude sensitivity, we compute cosine similarity and Pearson correlation between the frequency components of predictions and ground truth. For trend preservation, we measure the dominant frequency overlap to assess how well each model captures underlying trends. The results listed below show that our method achieves better performance in both magnitude and trend measures, indicating the effectiveness of our method in frequency learning.
>
> ||||S$^2$TS-LLM(Ours)|FEDformer|FiLM|
> |:-:|:-:|:-:|:-:|:-:|:-:|
> |ETTh1|Magtitude Test|Cosine Similarity↑|0.926|0.919|0.912|
> |||Pearson Correlation↑|0.878|0.865|0.857|
> ||Trend Test|Dominant Frequency Overlap↑|0.523|0.503|0.507|
> |ETTh2|Magtitude Test|Cosine Similarity↑|0.987|0.985|0.982|
> |||Pearson Correlation↑|0.985|0.982|0.979|
> ||Trend Test|Dominant Frequency Overlap↑|0.851|0.844|0.846|
>
> Based on the above results and the strong performance of our method across 23 datasets (as shown in Section 4), we believe that our method retains good generalization ability in real-world applications.
>
> > [1]T. Zhou, et al. Fedformer: Frequency enhanced decomposed transformer for long-term series forecasting. ICML
> [2]T. Zhou, et al. FiLM: Frequency improved Legendre Memory Model for Long-term Time Series Forecasting. NIPS
>
> ## W3:GPT-2 Gets Per-set LR Sweeps While Classic Models Use Canned Configs
>
> A: As described in Appendix B.3, our method follows the standard setup of existing time-series LLM methods [3][4][5] by fine-tuning the LLM with a fixed and small learning rate (e.g., 0.0001) for time-series analysis.
>
> > [3]T. Zhou, et al. One fits all: Power general time series analysis by pretrained lm. NIPS
> [4]M. Jin, et al. Time-llm: Time series forecasting by reprogramming large language models. ICLR
> [5]Z. J. Pan, et al. s2ip-llm: Semantic space informed prompt learning with llm for time series forecasting. ICML
>
> ## Q1:Contextual Segmentation vs. Long-Context Tricks
>
> A: We clarify that our segmentation and long-context modeling (the position encoding such as relative positions in RoPE) are two distinct components to address different problems: the former is an LLM-independent module that captures temporal dependencies from time series (as shown in Fig. 3), while the latter is part of the LLM used to model contextual relationships among input tokens. By default, if no inputs' indices are provided, the LLM automatically generates sequential indices (e.g., $P = [0,...,n-1]$) for inputs and pass them into the position encoding module to learn contextual relationships. Our method instead creates grouped indices $P$ and feeds them to the LLM. To illustrate why our segmentation works, we can consider treating each time-series patch as a character, such as $[x_0, x_1, x_2, x_3, x_4, x_5] =$ ["c", "a", "t", "d", "o", "g"]. A standard LLM would assign sequential indices [0,1,2,3,4,5], treating each patch independently. In contrast, our segmentation assigns grouped indices [0,0,0,1,1,1], effectively grouping $x_0$ to $x_2$ as "cat" and $x_3$ to $x_5$ as "dog". This narrows the structural gap between text and time series, allowing the LLM to better capture trend continuity in time series, without compromising its ability to model long-context dependencies.
>
> ## Q2:Effectiveness of Time-Linguistic Fusion
>
> A: To clarify, the proposed time-linguistic fusion (TLF) is designed in a simple yet effective way by combining cosine similarity with a gating mechanism. Specifically, the design of TLF is inspired by the Gated TCN module in GraphWavelet[6], which uses a gating mechanism of the form: $h = g(\Theta \star \mathcal{X} + b) \odot \sigma (\Theta \star \mathcal{X} + c)$. The sigmoid function $\sigma(\cdot)$ controls the proportion of information from $\mathcal{X}$ that is propagated to the next layer. In a similar spirit, we apply a sigmoid function to the frequency-domain output, generating a gating signal that controls the retention or suppression of time-series information during fusion. This gating is then complemented by a cosine similarity measure. It refines the fusion process by assigning higher weights to time-domain information that are more closely aligned with frequency-domain semantics.
>
> Below, we compare the proposed TLF with several other fusion techniques, including bilinear fusion ($z = x^T W y$), gate fusion ($z = \alpha x + (1-\alpha)y$), and attention-based fusion. The results show that TLF achieves the best performance, yielding lower prediction errors than the alternatives.
>
> |||TLF(Ours)|Bilinear|Gate|Att|
> |:-:|:-:|:-:|:-:|:-:|:-:|
> |ETTh1|Length|MSE↓, MAE↓|MSE↓, MAE↓|MSE↓, MAE↓|MSE↓, MAE↓|
> ||96|0.355, 0.387|0.359, 0.389|0.367, 0.396|0.369, 0.397|
> ||192|0.399, 0.419|0.412, 0.423|0.402, 0.422|0.401, 0.421|
> ||336|0.402, 0.432|0.439, 0.441|0.423, 0.443|0.446, 0.463|
> ||720|0.463, 0.478|0.462, 0.473|0.482, 0.485|0.502, 0.497|
> ||AVG|0.405, 0.429|0.418, 0.431|0.418, 0.436|0.430, 0.444|
> |ETTh2|96|0.260, 0.329|0.276, 0.335|0.273, 0.334|0.291, 0.345|
> ||192|0.312, 0.363|0.338, 0.375|0.335, 0.377|0.340, 0.379|
> ||336|0.322, 0.380|0.336, 0.385|0.333, 0.386|0.348, 0.398|
> ||720|0.392, 0.430|0.393, 0.428|0.400, 0.432|0.406, 0.439|
> ||AVG|0.322, 0.376|0.336, 0.381|0.335, 0.382|0.346, 0.390|
>
> > [6]Z. Wu, et al. Graph WaveNet for Deep Spatial-Temporal Graph Modeling. IJCAI
>
> ## Q3:Lack of Justification for GPT-2's Benefit from ASCII Symbols
>
> A: The effectiveness of ASCII symbols lies in that different combinations of ASCII symbols can reflect different distributional patterns of the time series and can be understood by LLMs. Specifically, we assign an ASCII symbol to each frequency component and encode the top-k components into a textual description of time series. When the resulting text is dominated by symbols representing low-frequency components, it indicates smooth and stable trends of time series; when high-frequency components dominate, it reflects more volatile changes of time series. Although ASCII symbols lacks semantic meaning in natural language, LLMs can, after fine-tuning, learn statistical patterns in ASCII sequences and interpret them as foreign-language tokens representing frequency [7]. As shown in the ablation study in Section 4.6, using ASCII symbols to represent frequency components achieves comparable performance to using natural-language words, demonstrating that ASCII symbols can effectively serve as interpretable tokens for LLMs.
>
> > [7]W. Su, et al. Do Large Language Models (Really) Need Statistical Foundations? arXiv

---

### Comment · Area_Chair_oHWa · 2025-08-05
**Feedback required**

Dear reviewer,

Please review the authors’ rebuttal and update your review with any clarifications, score adjustments, or remaining concerns. Your timely feedback helps us finalize decisions. Thank you!

--AC

---

### Decision · Program_Chairs · 2025-09-17

**Decision:**

Accept (poster)

**Comment:**

This paper tries to bridge time series and language with spectral symbolization and contextual segmentation. The idea is creative and the experiments are pretty broad, which most reviewers liked. At the same time, the novelty isn’t huge, since spectral ideas have been tried before, and things like the ASCII symbols and fusion design didn’t fully convince everyone. The rebuttal helps -- the authors added ablations, extra comparisons (FSCA, TimeCMA), and clarified some design choices, which made the work look more solid. While one reviewer stayed skeptical, the others leaned positive after the response.

Overall, I see this as a decent contribution, not perfect but worth accepting. Given the tough competition this year, I’d mark it as a weak accept.